# SPECTRAL DECOMPOSITION REPRESENTATION FOR REINFORCEMENT LEARNING

**Tongzheng Ren**[1,2,*]   **Tianjun Zhang**[1,3,*]   **Lisa Lee**[1]   **Joseph Gonzalez**[3]
**Dale Schuurmans**[1,4]   **Bo Dai**[1,5]
[1]Google Research, Brain Team [2]UT Austin [3]UC Berkeley [4]University of Alberta
[5]Georgia Tech
tongzheng@utexas.edu, tianjunz@berkeley.edu, bodai@google.com

## ABSTRACT

Representation learning often plays a critical role in avoiding the curse of dimensionality in reinforcement learning. A representative class of algorithms exploits spectral decomposition of the stochastic transition dynamics to construct representations that enjoy strong theoretical properties in idealized settings. However, current spectral methods suffer from limited applicability because they are constructed for state-only aggregation and are derived from a policy-dependent transition kernel, without considering the issue of exploration. To address these issues, we propose an alternative spectral method, *Spectral Decomposition Representation* (SPEDER), that extracts a *state-action* abstraction from the dynamics without inducing spurious dependence on the data collection policy, while also balancing the exploration-versus-exploitation trade-off during learning. A theoretical analysis establishes the sample efficiency of the proposed algorithm in both the online and offline settings. In addition, an experimental investigation demonstrates superior performance over current state-of-the-art algorithms across several RL benchmarks.

## 1 INTRODUCTION

Reinforcement learning (RL) seeks to learn an optimal sequential decision making strategy by interacting with an unknown environment, usually modeled by a Markov decision process (MDP). For MDPs with finite states and actions, RL can be performed in a sample efficint and computationally efficient way; however, for large or infinite state spaces both the sample and computational complexity increase dramatically. Representation learning is therefore a major tool to combat the implicit curse of dimensionality in such spaces, contributing to several empirical successes in deep RL, where policies and value functions are represented as deep neural networks and trained end-to-end (Mnih et al., 2015; Levine et al., 2016; Silver et al., 2017; Bellemare et al., 2020). However, an inappropriate representation can introduce approximation error that grows exponentially in the horizon (Du et al., 2019b), or induce redundant solutions to the Bellman constraints with large generalization error (Xiao et al., 2021). Consequently, ensuring the quality of representation learning has become an increasingly important consideration in deep RL.

In prior work, many methods have been proposed to ensure alternative properties of a learned representation, such as reconstruction (Watter et al., 2015), bi-simulation (Gelada et al., 2019; Zhang et al., 2020), and contrastive learning (Zhang et al., 2022a; Qiu et al., 2022; Nachum & Yang, 2021). Among these methods, a family of representation learning algorithms has focused on constructing features by exploiting the spectral decomposition of different transition operators, including successor features (Dayan, 1993; Machado et al., 2018), proto-value functions (Mahadevan & Maggioni, 2007; Wu et al., 2018), spectral state aggregation (Duan et al., 2019; Zhang & Wang, 2019), and Krylov bases (Petrik, 2007; Parr et al., 2008). Although these algorithms initially appear distinct, they all essentially factorize a variant of the transition kernel. The most attractive property of such representations is that the value function can be *linearly* represented in the learned features, thereby reducing the complexity of subsequent planning. Moreover, spectral representations are compatible with deep neural networks (Barreto et al., 2017), which makes them easily applicable to optimal policy learning (Kulkarni et al., 2016b) in deep RL.

---

*Equal Contribution.

However, despite their elegance and desirable properties, current spectral representation algorithms exhibit several drawbacks. One drawback is that current methods generate *state-only* features, which are heavily influenced by the behavior policy and can fail to generalize well to alternative polices. Moreover, most existing spectral representation learning algorithms omit the intimate coupling between representation learning and exploration, and instead learn the representation from a *pre-collected static dataset*. This is problematic as effective exploration depends on having a good representation, while learning the representation requires comprehensively-covered experiences—failing to properly manage this interaction can lead to fundamentally sample-inefficient data collection (Xiao et al., 2022). These limitations lead to suboptimal features and limited empirical performance.

In this paper, we address these important but largely ignored issues, and provide a novel spectral representation learning method that generates *policy-independent* features that provably manage the delicate balance between *exploration and exploitation*. In summary:

- We provide a spectral decomposition view of several current representation learning methods, and identify the cause of spurious dependencies in state-only spectral features (Section 2.2).
- We develop a novel model-free objective, *Spectral Decomposition Representation (SPEDER)*, that factorizes the policy-independent transition kernel to eliminate policy-induced dependencies, while revealing the connection between model-free and model-based representation learning (Section 3).
- We provide algorithms that implement the principles of optimism and pessimism in the face of uncertainty using the SPEDER features for online and offline RL (Section 3.1), and equip behavior cloning with SPEDER for imitation learning (Section 3.2).
- We analyze the sample complexity of SPEDER in both the online and offline settings, to justify the achieved balance between exploration versus exploitation (Section 4).
- We demonstrate that SPEDER outperforms state-of-the-art model-based and model-free RL algorithms on several benchmarks (Section 6).

## 2 PRELIMINARIES

In this section, we briefly introduce Markov Decision Processes (MDPs) with a low-rank structure, and reveal the spectral decomposition view of several representation learning algorithms, which motivates our new spectral representation learning algorithm.

### 2.1 LOW-RANK MARKOV DECISION PROCESSES

Markov Decision Processes (MDPs) are a standard sequential decision-making model for RL, and can be described as a tuple $\mathcal{M} = (\mathcal{S}, \mathcal{A}, r, P, \rho, \gamma)$, where $\mathcal{S}$ is the state space, $\mathcal{A}$ is the action space, $r : \mathcal{S} \times \mathcal{A} \to [0, 1]$ is the reward function, $P : \mathcal{S} \times \mathcal{A} \to \Delta(\mathcal{S})$ is the transition operator with $\Delta(\mathcal{S})$ as the family of distributions over $\mathcal{S}$, $\rho \in \Delta(\mathcal{S})$ is the initial distribution and $\gamma \in (0, 1)$ is the discount factor. The goal of RL is to find a policy $\pi : \mathcal{S} \to \Delta(\mathcal{A})$ that maximizes the cumulative discounted reward $V_{P,r}^{\pi} := \mathbb{E}_{s_0 \sim \rho, \pi} \left[ \sum_{i=0}^{\infty} \gamma^i r(s_i, a_i) | s_0 \right]$ by interacting with the MDP. The value function is defined as $V_{P,r}^{\pi}(s) = \mathbb{E}_{\pi} \left[ \sum_{i=0}^{\infty} \gamma^i r(s_i, a_i) | s_0 = s \right]$, and the action-value function is $Q_{P,r}^{\pi}(s, a) = \mathbb{E}_{\pi} \left[ \sum_{i=0}^{\infty} \gamma^i r(s_i, a_i) | s_0 = s, a_0 = a \right]$. These definitions imply the following recursive relationships:

$$V_{P,r}^{\pi}(s) = \mathbb{E}_{\pi} \left[ Q_{P,r}^{\pi}(s, a) \right], \quad Q_{P,r}^{\pi}(s, a) = r(s, a) + \gamma \mathbb{E}_P \left[ V_{P,r}^{\pi}(s') \right].$$

We additionally define the state visitation distribution induced by a policy $\pi$ as $d_P^{\pi}(s) = (1 - \gamma) \mathbb{E}_{s_0 \sim \rho, \pi} \mathbb{E} \left[ \sum_{t=0}^{\infty} \gamma^t \mathbf{1}(s_t = s) | s_0 \right]$, where $\mathbf{1}(\cdot)$ is the indicator function.

When $|\mathcal{S}|$ and $|\mathcal{A}|$ are finite, there exist sample-efficient algorithms that find the optimal policy by maintaining an estimate of $P$ or $Q_{P,r}^{\pi}$ (Azar et al., 2017; Jin et al., 2018). However, such methods cannot be scaled up when $|\mathcal{S}|$ and $|\mathcal{A}|$ are extremely large or infinite. In such cases, function approximation is needed to exploit the structure of the MDP while avoiding explicit dependence on $|\mathcal{S}|$ and $|\mathcal{A}|$. The low rank MDP is one of the most prominent structures that allows for simple yet effective function approximation in MDPs, which is based on the following spectral structural assumption on $P$ and $r$:

**Assumption 1** (Low Rank MDP, (Jin et al., 2020; Agarwal et al., 2020)). *An MDP $\mathcal{M}$ is a low rank MDP if there exists a low rank spectral decomposition of the transition kernel $P(s'|s, a)$, such that*

$$P(s'|s, a) = \langle \phi(s, a), \mu(s') \rangle, \quad r(s, a) = \langle \phi(s, a), \theta_r \rangle, \tag{1}$$

with two spectral maps $\phi : \mathcal{S} \times \mathcal{A} \to \mathbb{R}^d$ and $\mu : \mathcal{S} \to \mathbb{R}^d$, and a vector $\theta_r \in \mathbb{R}^d$. The $\phi$ and $\mu$ also satisfy the following normalization conditions:

$$\forall (s,a), \ \|\phi(s,a)\|_2 \leqslant 1, \ \|\theta_r\|_2 \leqslant \sqrt{d}, \forall g : \mathcal{S} \to \mathbb{R}, \ \|g\|_{L_\infty} \leqslant 1 \,, \left\|\int_{\mathcal{S}} \mu(s')g(s')ds'\right\|_2 \leqslant \sqrt{d}. \quad (2)$$

The low rank MDP allows for a linear representation of $Q_{P,r}^\pi$ for any *arbitrary policy* $\pi$, since

$$Q_{P,r}^\pi(s,a) = r(s,a) + \gamma \int V_{P,r}^\pi(s) P(s'|s,a)ds' = \left\langle \phi(s,a), \theta_r + \gamma \int V_{P,r}^\pi(s')\mu(s')ds' \right\rangle. \quad (3)$$

Hence, we can provably perform computationally-efficient planning and sample-efficient exploration in a low-rank MDP given $\phi(s,a)$, as shown in (Jin et al., 2020). However, $\phi(s,a)$ is generally unknown to the reinforcement learning algorithm, and must be learned via representation learning to leverage the structure of low rank MDPs.

## 2.2 SPECTRAL FRAMEWORK FOR REPRESENTATION LEARNING

Representation learning based on a spectral decomposition of the transition dynamics was investigated as early as Dayan (1993), although the explicit study began with Mahadevan & Maggioni (2007), which constructed features via eigenfunctions from Laplacians of the transitions. This inspired a series of subsequent work on spectral decomposition representations, including the Krylov basis (Petrik, 2007; Parr et al., 2008), continuous Laplacian Wu et al. (2018), and spectral state aggregation (Duan et al., 2019; Zhang & Wang, 2019). We summarize these algorithms in Table 1 to reveal their commonality from a unified perspective, which motivates the development of a new algorithm. A similar summary has also been provided in (Ghosh & Bellemare, 2020).

These existing spectral representation methods construct features based on the spectral space of the state transition probability $P^\pi(s'|s) = \int P(s'|s,a)\pi(a|s)da$, induced by some policy $\pi$. Such a transition operator introduces inter-state dependency from the specific $\pi$, and thus injects an inductive bias into the state-only spectral representation, resulting in features that might not be generalizable to other policies. To make the state-feature generalizable, some work has resorted incorporating a linear action model (*e.g.* Yao & Szepesvári, 2012; Gehring et al., 2018) where action information is stored in the linear weights. However, this work requires known state-features, and it is not clear how to combine the linear action model with the spectral feature framework. Moreover, these existing spectral representation methods completely ignore the problem of exploration, which affects the composition of the dataset for representation learning and is conversely affected by the learned representation during the data collection procedure. These drawbacks have limited the performance of spectral representations in practice.

Table 1: A unified spectral decomposition view of existing related representations. Here, $r$ denotes the reward function, $\Lambda$ denotes some diagonal reweighting operator, and $P^\pi(s'|s) = \int P(s'|s,a)\pi(a|s)da$.

| Representation | Decomposed Dynamics |
|---|---|
| Successor Feature | $\texttt{svd}\left((I - \gamma P^\pi)^{-1}\right)$ |
| Proto-Value Function | $\texttt{eig}\left(\Lambda P^\pi + (P^\pi)^\top \Lambda\right)$ |
| Krylov Basis | $\{(P^\pi)^i r\}_{i=1}^k$ |
| Spectral State-Aggregation | $\texttt{svd}(P^\pi)$ |

## 3 SPECTRAL DECOMPOSITION REPRESENTATION LEARNING

To address these issues, we provide a novel spectral representation learning method, which we call *SPEctral DEcomposition Representation (SPEDER)*. SPEDER is compatible with stochastic gradient updates, and is therefore naturally applicable to general practical settings. We will show that SPEDER can be easily combined with the principle of optimism in the face of uncertainty to obtain sample efficient online exploration, and can also be leveraged to perform latent behavioral cloning.

As discussed in Section 2, the fundamental cause of the spurious dependence in state-only spectral features arises from the state transition operator $P^\pi(s'|s)$, which introduces inter-state dependence induced by a specific behavior policy $\pi$. To resolve this issue, we extract the spectral feature from $P(s'|s,a)$ alone, which is *invariant* to the policy, thereby resulting in a more stable spectral representation.

Assume we are given a set of observations $\{(s_i, a_i, s_i')\}_{i=1}^n$ sampled from $\rho_0(s,a) \times P(s'|s,a)$, and want to learn spectral features $\phi(s,a) \in \mathbb{R}^d$ and $\mu(s') \in \mathbb{R}^d$, which are produced by function approximators like deep neural nets, such that:

$$P(s'|s,a) \approx \phi(s,a)^\top \mu(s'). \quad (4)$$

Such a representation allows for a simple linear parameterization of $Q$ for any policy $\pi$, making the planning step efficient, as discussed in section 2.1. Based on (4), one can exploit density estimation techniques, *e.g.*, maximum likelihood estimation (MLE), to estimate $\phi$ and $\mu$:

$$\left(\widehat{\phi}, \widehat{\mu}\right) = \underset{\phi \in \Phi, \mu \in \Psi}{\arg\max} \frac{1}{n} \sum_{i=1}^{n} \log \phi(s_i, a_i)^\top \mu(s_i') - \log Z(s, a), \tag{5}$$

where $Z(s, a) = \int_{\mathcal{S}} \phi(s, a)^\top \mu(s') \, ds'$. In fact, (Agarwal et al., 2020; Uehara et al., 2022) provide rigorous theoretical guarantees when a computation oracle for solving (5) is provided. However, such a computation for MLE is highly nontrivial. Meanwhile, the MLE (5) is invariant to the scale of $\phi$ and $\mu$; that is, if $(\phi, \mu)$ is a solution of (5), then $(c_1\phi, c_2\mu)$ is also a solution of (5) for any $c_1, c_2 > 0$. Hence, we generally do not have $Z(s, a) = 1$ for any $(s, a)$, and we can only use $P(s'|s, a) = \left(\frac{\phi(s,a)}{Z(s,a)}\right)^\top \mu(s')$. Therefore, we need to use $\widetilde{\phi}(s, a) := \frac{\phi(s,a)}{Z(s,a)}$ to linearly represent the $Q$-function, which incurs an extra estimation requirement for $Z(s, a)$.

Recall that the pair $\phi(s, a)$ and $\mu(s')$ actually form the subspace of transition operator $P(s'|s, a)$, so instead of MLE for the factorization of $P(s'|s, a)$, we can directly apply singular value decomposition (SVD) to the transition operator to bypass the computation difficulties in MLE. Specifically, the SVD of transition operator can be formulated as

$$\max_{\mathbb{E}[\phi\phi^\top]=I_d} \left\| \mathbb{E}_{\rho_0} \left[ P(s'|s, a)\phi(s, a) \right] \right\|_2^2 \tag{6}$$

$$= \max_{\mathbb{E}[\phi\phi^\top]=I_d/d} \max_{\mu} 2\text{Trace}\left( \mathbb{E}_{\rho_0} \left[ \int \mu(s')P(s'|s, a)\phi(s, a)^\top ds' \right] \right) - 1/d \int \mu(s')^\top \mu(s') ds'$$

$$= \max_{\mathbb{E}[\phi\phi^\top]=I_d/d, \mu'} 2\mathbb{E}_{\rho_0 \times P} \left[ \phi(s, a)^\top \mu'(s') p(s') \right] - \mathbb{E}_p \left[ p(s')\mu'(s')^\top \mu'(s') \right] /d, \tag{7}$$

where $\|\cdot\|_2$ denotes the $L_2(\mu)$ norm where $\mu$ denotes the Lebesgue measure for continuous case and counting measure for discrete case, the second equality comes from the Fenchel duality of $\|\cdot\|_2^2$ with up-scaling of $\mu$ by $\sqrt{d}$, and the third equality comes from reparameterization $\mu(s') = p(s')\mu'(s')$ with some parametrized probability measure $p(s')$ supported on the state space $\mathcal{S}$.

As (7) can be approximated with the samples, it can be solved via stochastic gradient updates, where the constraint is handled via the penalty method as in (Wu et al., 2018). This algorithm starkly contrasts with existing policy-dependent methods for spectral features via explicit eigendecomposition of state transition matrices (Mahadevan & Maggioni, 2007; Machado et al., 2017; 2018).

**Remark (equivalent model-based view of (7)):** We emphasize that the SVD variational formulation in (6) is model-free, according to the categorization in Modi et al. (2021), without explicit modeling of $\mu$. Here, $\mu$ is only introduced only for tractability. This reveals an interesting model-based perspective on representation learning.

We draw inspiration from spectral conditional density estimation (Grünewälder et al., 2012):

$$\min_{\phi, \mu} \mathbb{E}_{(s,a)\sim\rho_0} \left\| P(\cdot|s, a) - \phi(s, a)^\top \mu(\cdot) \right\|_2^2. \tag{8}$$

This objective (8) has a unique global minimum, $\phi(s, a)^\top \mu(s') = P(s'|s, a)$, thus it can be used as an alternative representation learning objective.

However, the objective (8) is still intractable when we only have access samples from $P$. To resolve the issue, we note that

$$L(\phi, \mu) := \mathbb{E}_{(s,a)\sim\rho_0} \left\| P(\cdot|s, a) - \phi(s, a)^\top \mu(\cdot) \right\|_2^2$$

$$= C - 2\mathbb{E}_{(s,a)\sim\rho_0, s'\sim P(s'|s,a)} \left[ \phi(s, a)^\top \mu(s') \right] + \mathbb{E}_{(s,a)\sim\rho_0} \left[ \int_{\mathcal{S}} \left( \phi(s, a)^\top \mu(s') \right)^2 ds' \right], \tag{9}$$

where $C = \mathbb{E}_{s,a\sim\rho_0} \left[ \int \left( P(s'|s, a) \right)^2 \right]$ is a problem-dependent constant. For the third term, we turn to an approximation method by reparameterization $\mu(s') = p(s')\mu'(s')$,

$$\mathbb{E}_{(s,a)\sim\rho_0} \left[ \int_{\mathcal{S}} \left( \phi(s, a)^\top \mu(s') \right)^2 ds' \right] = \text{Trace}\left( \mathbb{E}_{(s,a)\sim\rho_0} \left[ \phi(s, a)\phi(s, a)^\top \right] \mathbb{E}_p \left[ p(s')\mu'(s')\mu'(s')^\top \right] \right).$$

---

**Algorithm 1** Online Exploration with SPEDER

---

1: **Input:** Regularizer $\lambda_n$, parameter $\alpha_n$, Model class $\mathcal{F} = \{(\phi, \mu) : \phi \in \Phi, \mu \in \Psi, \}$, Iteration $N$
2: Initialize $\pi_0(\cdot \mid s)$ to be uniform; set $\mathcal{D}_0 = \emptyset, \mathcal{D}'_0 = \emptyset$
3: **for** episode $n = 1, \cdots, N$ **do**
4:      Collect the transition $(s, a, s', a', \tilde{s})$ where $s \sim d_{P^\star}^{\pi_{n-1}}$, $a \sim \mathcal{U}(\mathcal{A})$, $s' \sim P^\star(\cdot|s,a)$,$a' \sim \mathcal{U}(\mathcal{A})$, $\tilde{s} \sim P^\star(\cdot|s', a')$, where $\mathcal{U}(\mathcal{A})$ denotes the uniform distribution on $\mathcal{A}$.
5:      $\mathcal{D}_n = \mathcal{D}_{n-1} \cup \{s, a, s'\}, \mathcal{D}'_n = \mathcal{D}'_{n-1} \cup \{s', a', \tilde{s}\}$.
6:      Learn representation $\widehat{\phi}(s, a)$ with $\mathcal{D}_n \cup \mathcal{D}'_n$ via equation 10 .
7:      Update the empirical covariance matrix
$$\widehat{\Sigma}_n = \sum_{s,a \in \mathcal{D}_n} \widehat{\phi}_n(s, a)\widehat{\phi}_n(s, a)^\top + \lambda_n I$$
8:      Set the exploration bonus $\widehat{b}_n(s, a) = \alpha_n \sqrt{\widehat{\phi}_n(s, a)^\top \widehat{\Sigma}_n^{-1} \widehat{\phi}_n(s, a)}$
9:      Update policy $\pi_n = \arg\max_\pi V_{\widehat{P}_n, r+\widehat{b}_n}^\pi$
10: **end for**
11: **Return** $\pi_1, \cdots, \pi_N$

---

Under the constraint that $\mathbb{E}_{s,a}[\phi(s, a)\phi(s, a)^\top] = I_d/d$, we have

$$\text{Trace}\left(\mathbb{E}_{(s,a) \sim \rho_0}\left[\phi(s, a)\phi(s, a)^\top\right]\mathbb{E}_p\left[p(s')\mu'(s')\mu'(s')^\top\right]\right) = \mathbb{E}_p\left[p(s')\mu'(s')^\top\mu'(s')\right]/d.$$

Hence, Equation 8 can be written equivalently as:

$$\min_{\phi, \mu'} -\mathbb{E}_{(s,a,s') \sim \rho_0 \times P}\left[\phi(s, a)^\top \mu'(s')p(s')\right] + \left(\mathbb{E}_{p(s')}\left[p(s')\mu'(s')^\top \mu'(s')\right]\right)/(2d)$$

$$\text{s.t. } \mathbb{E}_{(s,a) \sim \rho_0}\left[\phi(s, a)\phi(s, a)^\top\right] = I_d/d, \quad (10)$$

which is exact as the dual form of the SVD in (7). Such an equivalence reveals an interesting connection between model-free and model-based representation learning, obtained through duality, which indicates that the spectral representation learned via SVD is implicitly minimizing the model error in $L_2$ norm. This connection paves the way for theoretical analysis.

### 3.1 ONLINE EXPLORATION AND OFFLINE POLICY OPTIMIZATION WITH SPEDER

Unlike existing spectral representation learning algorithms, where the features are learned based on a pre-collected static dataset, we can use SPEDER to perform sample efficient online exploration. In Algorithm 1, we show how to use the representation obtained from the solution to (10) to perform sample efficient online exploration under the principle of optimism in the face of uncertainty. Central to the algorithm is the newly proposed representation learning procedure (Line 6 in Algorithm 1), which learns the representation $\widehat{\phi}(s, a)$ and the model $\widehat{P}(s'|s, a) = \widehat{\phi}(s, a)^\top \widehat{\mu}(s)$ with adaptively collected exploratory data. After recovering the representation, we use the standard elliptical potential (Jin et al., 2020; Uehara et al., 2022) as the bonus (Line 8 in Algorithm 1) to enforce exploration. We then plan using the learned model $\widehat{P}_n$ with the reward bonus $\widehat{b}_n$ to obtain a new policy that is used to collect additional exploratory data. These procedures iterate, comprising Algorithm 1.

SPEDER can also be combined with the pessimism principle to perform sample efficient offline policy optimization. Unlike the online setting where we enforce exploration by adding a bonus to the reward, we now *subtract* the elliptical potential from the reward to avoid risky behavior. For completeness, we include the algorithm for offline policy optimization in Appendix C.

**On the requirements of $\widehat{P}_n$.** As we need to plan with the learned model $\widehat{P}_n$, we generally require $\widehat{P}_n$ to be a valid transition kernel, but the representation learning objective (10) does not explicitly enforce this. Therefore in our implementations, we use the data from the replay buffer collected during the past executions to perform planning. We can also enforce that $\widehat{P}_n$ is a valid probability by adding the following additional regularization term Ma & Collins (2018):

$$\mathbb{E}_{(s,a)}\left[\left(\log \int_{\mathcal{S}} \phi(s, a)^\top \mu'(s')p(s')ds'\right)^2\right], \quad (11)$$

which can be approximated with samples from $p(s')$. Obviously, the regularization is non-negative and achieves zero when $\int_{\mathcal{S}} \phi(s, a)^\top \mu'(s')p(s')ds' = 1$.

**Practical Implementation**  We parameterize $\phi(s, a)$ and $\mu'(s')$ as separate MLP networks, and train them by optimizing objective (10). Instead of using a linear $Q$ on top of $\phi(s, a)$, as suggested by the low-rank MDP, we parameterize the critic network as a two-layer MLP on top of the learned representation $\phi(s, a)$ to support the nonlinear exploration bonus and entropy regularization. Unlike other representation learning methods in RL, we do not backpropagate the gradient from TD-learning to the representation network $\phi(s, a)$. To train the policy, we use the Soft Actor-Critic (SAC) algorithm (Haarnoja et al., 2018), and alternate between policy optimization and critic training.

## 3.2    SPECTRAL REPRESENTATION FOR LATENT BEHAVIORAL CLONING

We additionally expand the use of the learned spectral representation $\phi(s, a)$ as skills for downstream imitation learning, which seeks to mimic a given set of expert demonstrations. Specifically, recall the correspondence between the max-entropy policy and $Q$-function, *i.e.*,

$$\pi_Q(a|s) := \frac{\exp(Q(s, a))}{\sum_{a \in \mathcal{A}} \exp(Q(s, a))} = \underset{\pi(\cdot|s) \in \Delta(\mathcal{A})}{\arg \max} \; \mathbb{E}_\pi \left[ Q(s, a) \right] + H(\pi), \tag{12}$$

where $H(\pi) := \sum_{a \in \mathcal{A}} \pi(a|s) \log \pi(a|s)$. Therefore, given a set of linear basis functions for $Q$, $\{\phi_i\}_{i=1}^d$, the we can construct the policy basis, or skill sets, based on $\phi$ according to (12), which induces the policy family $\pi_w(a|s) \propto \exp(w^\top \phi(s, a))$. We emphasize that the policy construction from the skills is *no longer linear*. This inspires us to use a latent variable composition to approximate policy construction, *i.e.*, $\pi(a|s) = \int \pi_\alpha(a|s, z) \pi_Z(z|s) dz$, with $z = \phi(s, a)$ to insert the learned representation. The policy decoder $\pi_\alpha : \mathcal{S} \times Z \to \Delta(\mathcal{A})$ and the policy encoder $\pi_Z : \mathcal{S} \to \Delta(Z)$ can be composed to form the final policy.

We assume access to a fixed set of expert transitions $\mathcal{D}^{\pi^*} = \{(s_t, a_t, s_{t+1}) : s_t \sim d_P^{\pi^*}, a_t \sim \pi^E(s_t), s_{t+1} \sim P(s' \mid s_t, a_t)\}$. In practice, while expert demonstrations can be expensive to acquire, non-expert data of interactions in the same environment can be more accessible to collect at scale, and provide additional information about the transition dynamics of the environment. We denote the offline transitions $\mathcal{D}^{\text{off}} = \{(s, a, s')\}$ from the same MDP, which is collected by a non-expert policy with suboptimal performance (*e.g.*, an exploratory policy). We follow latent behavioral cloning (Yang et al., 2021; Yang & Nachum, 2021) where learning is separated into a pre-training phase, where a representation $\phi : \mathcal{S} \times \mathcal{A} \to Z$ and a policy decoder $\pi_\alpha : \mathcal{S} \times Z \to \Delta(\mathcal{A})$ are learned on the basis of the suboptimal dataset $\mathcal{D}^{\text{off}}$, and a downstream imitation phase that learns a latent policy $\pi_Z : \mathcal{S} \to \Delta(Z)$ using the expert dataset $\mathcal{D}^{\pi^*}$. With SPEDER, we perform latent behavior cloning as follows:

1. **Pretraining Phase:** We pre-train $\phi(s, a)$ and $\mu(s')$ on $\mathcal{D}^{\text{off}}$ by minimizing the objective (10). Additionally, we train a policy decoder $\pi_\alpha(a \mid s, \phi(s, a))$ that maps latent action representations to actions in the original action space, by minimizing the action decoding error:

$$\mathbb{E}_{s \sim d_P^{\text{off}}} \left[ -\log \pi_\alpha(a \mid s, \phi(s, a)) \right]$$

2. **Downstream Imitation Phase:** We train a latent policy $\pi_Z : S \to \Delta(Z)$ by minimizing the latent behavioral cloning error:

$$\mathbb{E}_{(s, a) \sim d_P^{\pi^*}} \left[ -\log \pi_Z(\phi(s, a) \mid s) \right]$$

At inference time, given the current state $s \in \mathcal{S}$, we sample a latent action representation $z \sim \pi_Z(s)$, then decode the action $a \sim \pi_\alpha(a \mid s, z)$.

## 4    THEORETICAL ANALYSIS

In this section, we establish generalization properties of the proposed representation learning algorithm, and provide sample complexity and error bounds when the proposed representation learning algorithm is applied to online exploration and offline policy optimization.

### 4.1    NON-ASYMPTOTIC GENERALIZATION BOUND

We first state a performance guarantee on the representation learned with the proposed objective.

**Theorem 1.** *Assume the size of candidate model class $|\mathcal{F}| < \infty$, $P \in \mathcal{F}$, and for any $\tilde{P} \in \mathcal{F}$, $\tilde{P}(s'|s, a) \leqslant C$ for all $(s, a, s')$. Given the dataset $\mathcal{D} := \{(s_i, a_i, s_i')\}_{i=1}^n$ where $(s_i, a_i) \sim \rho_0$,*

$s_i' \sim P(\cdot|s_i, a_i)$, *the estimator* $\widehat{P}$ *obtained by empirical surrogate of* (9) *satisfies the following inequality with probability at least* $1 - \delta$:

$$\mathbb{E}_{(s,a)\sim\rho_0}\|P(\cdot|s,a) - \widehat{P}(\cdot|s,a)\|_2^2 \leqslant \frac{C' \log |\mathcal{F}|/\delta}{n}, \tag{13}$$

*where $C'$ is a constant that only depends on $C$, which we will omit in the following analysis.*

Note that the *i.i.d.* data assumption can be relaxed to an assumption that the data generating process is a martingale process. This is essential for proving the sample complexity of online exploration, as the data are collected in an adaptive manner. The proofs are deferred to Appendix D.1.

### 4.2 Sample Complexities of Online Exploration and Offline Policy Optimization

Next, we establish sample complexities for the online exploration and offline policy optimization problems. We still assume $P \in \mathcal{F}$. As the generalization bound equation 13 only guarantees the expected $L_2$ distance, we need to make the following additional assumptions on the representation and reward:

**Assumption 2** (Representation Normalization). $\forall \phi \in \Phi$, *we have* $\int_{\mathcal{S}} \left( \int_{\mathcal{A}} \|\phi(s,a)\|_2 \, \mathrm{d}a \right)^2 \mathrm{d}s \leqslant d$.

**Assumption 3** (Reward Normalization). $\int_{\mathcal{S}} \left( \int_{\mathcal{A}} r(s,a) \, \mathrm{d}a \right)^2 \mathrm{d}s \leqslant d$, *where $r$ is the reward function.*

A simple example that satisfies both Assumption 2 and 3 is a tabular MDP with features $\phi(s,a)$ forming the canonical basis in $\mathbb{R}^{|\mathcal{S}||\mathcal{A}|}$. In this case, we have $d = |\mathcal{S}||\mathcal{A}|$, hence Assumption 2 naturally holds. Furthermore, since $r(s,a) \in [0,1]$, it is also straightforward to verify that Assumption 3 holds for a tabular MDP. Such an assumption can also be satisfied for a continuous state space where the volume of the state space satisfies $\mu(\mathcal{S}) \leqslant \frac{d}{|\mathcal{A}|}$. Since we need to plan on $\widehat{P}$, we also assume $\widehat{P}$ is a valid transition kernel. With Assumptions 2 and 3 in hand, we are now ready to provide the sample complexities of online exploration and offline policy optimization. The proofs are deferred to Appendix D.2 and D.3.

**Theorem 2** (PAC Guarantee for Online Exploration). *Assume* $|\mathcal{A}| < \infty$. *After interacting with the environment for* $N = \widetilde{\Theta}\left( \frac{d^4 |\mathcal{A}|^2}{(1-\gamma)^6 \epsilon^2} \right)$ *episodes, where* $\widetilde{\Theta}$ *omits* log*-factors, we obtain a policy* $\pi$ *s.t.*

$$V_{P,r}^{\pi^*} - V_{P,r}^{\pi} \leqslant \epsilon$$

*with high probability, where $\pi^*$ is the optimal policy. Furthermore, note that, we can obtain a sample from the state visitation distribution $d_P^\pi$ via terminating with probability $1 - \gamma$ for each step. Hence, for each episode, we can terminate within $\widetilde{\Theta}(1/(1-\gamma))$ steps with high probability.*

**Theorem 3** (PAC Guarantee for Offline Policy Optimization). *Let* $\omega = \min_{s,a} \frac{1}{\pi_b(a|s)}$ *where $\pi_b$ is the behavior policy. With probability $1 - \delta$, for all baseline policies $\pi$ including history-dependent non-Markovian policies, we have that*

$$V_{P,r}^{\pi} - V_{P,r}^{\widehat{\pi}} \lesssim \sqrt{\frac{\omega^2 d^4 C_\pi^* \log(|\mathcal{F}|/\delta)}{(1-\gamma)^6}},$$

*where $C_\pi^*$ is the relative conditional number under $\phi^*$ which measures the quality of the offline data:*

$$C_\pi^* := \sup_{x \in \mathbb{R}} \frac{x^\top \mathbb{E}_{(s,a)\sim d_P^\pi}[\phi^*(s,a)\phi^*(s,a)^\top]x}{x^\top \mathbb{E}_{(s,a)\sim\rho_b}[\phi^*(s,a)\phi^*(s,a)^\top]x}.$$

## 5 Related Work

Aside from the family of spectral decomposition representation methods reviewed in Section 2, there have been many attempts to provide **algorithmic** representation learning algorithms for RL in different problem settings. Learning *action* representations, or abstractions, such as temporally-extended skills, has been a long-standing focus of hierarchical RL (Dieterich et al., 1998; Sutton et al., 1999; Kulkarni et al., 2016a; Nachum et al., 2018) for solving temporally-extended tasks. Recently, many algorithms have been proposed for online unsupervised skill discovery, which can reduce the cost of exploration and sample complexity of online RL algorithms. A class of methods extract temporally-extended skills by maximizing a mutual information objective (Eysenbach et al., 2018; Sharma et al., 2019; Lynch et al., 2020) or minimizing divergences (Lee et al., 2019). Unsupervised skill discovery has

been also studied in offline settings, where the goal is to pre-train useful skill representations from offline trajectories, in order to accelerate learning on downstream RL tasks (Yang & Nachum, 2021). Such methods include OPAL (Ajay et al., 2020), SPiRL (Pertsch et al., 2020), and SkiLD (Pertsch et al., 2021), which exploit a latent variable model with an autoencoder for skills acquisition; and PARROT (Singh et al., 2020), which learns a behavior prior with flow-based models. Another offline representation learning algorithm, TRAIL (Yang et al., 2021), uses a contrastive implementation of the MLE for an energy-based model to learn state-action features. These algorithms achieve empirical improvements in different problem settings, such as imitation learning, policy transfer, etc. However, as far as we know, the coupling between exploration and representation learning has not been well handled, and there is no rigorous characterization yet for these algorithms.

Another line of research focuses on **theoretically** guaranteed representation learning in RL, either by limiting the flexibility of the models or by ignoring the practical issue of computational cost. For example, (Du et al., 2019a; Misra et al., 2020) considered representation learning in block MDPs, where the representation can be learned via regression. However, the corresponding representation ability is *exponentially weaker* than low-rank MDPs (Agarwal et al., 2020). Ren et al. (2021) exploited representation from arbitrary dynamics models, but restricted the noise model to be Gaussian. On the other hand, (Agarwal et al., 2020; Modi et al., 2021; Uehara et al., 2022; Zhang et al., 2022b; Chen et al., 2022) provably extracted spectral features in low-rank MDPs with exploration, but these methods rely on a strong computation oracle, which is difficult to implement in practice.

In contrast, SPEDER enjoys both theoretical and empirical advantages. We provide a tractable surrogate with an efficient algorithm for spectral feature learning with exploration in low-rank MDPs. We have established its sample complexity and next demonstrate its superior empirical performance.

## 6 EXPERIMENTS

We evaluate SPEDER on the dense-reward MuJoCo tasks (Brockman et al., 2016) and sparse-reward DeepMind Control Suite tasks (Tassa et al., 2018). In MuJoCo tasks, we compare with model-based (*e.g.*, PETS (Chua et al., 2018), ME-TRPO (Kurutach et al., 2018)) and model-free baselines (*e.g.*, SAC (Haarnoja et al., 2018), PPO (Schulman et al., 2017)), showing strong performance compared to SoTA RL algorithms. In particular, we find that in the sparse reward DeepMind Control tasks, the optimistic SPEDER significantly outperforms the SoTA model-free RL algorithms. We also evaluate the method on offline behavioral cloning tasks in the AntMaze environment using the D4RL benchmark (Fu et al., 2020), and show comparable results to state-of-the-art representation learning methods. Additional details about the experiment setup are described in Appendix F.

### 6.1 ONLINE PERFORMANCE WITH THE SPECTRAL REPRESENTATION

We evaluate the proposed algorithm on the dense-reward MuJoCo benchmark from MBBL (Wang et al., 2019). We compare SPEDER with several model-based RL baselines (PETS (Chua et al., 2018), ME-TRPO (Kurutach et al., 2018)) and SoTA model-free RL baselines (SAC (Haarnoja et al., 2018), PPO (Schulman et al., 2017)). As a standard evaluation protocol in MBBL, we ran all the algorithms for 200K environment steps. The results are averaged across four random seeds with window size 20K.

In Table 2, we show that SPEDER achieves SoTA results among all model-based RL algorithms and significantly improves the prior baselines. We also compare the algorithm with the SoTA model-free RL method SAC. The proposed method achieves comparable or better performance in most of the tasks. Lastly, compared to two representation learning baselines (Deep SF (Kulkarni et al., 2016b) and SPEDE (Ren et al., 2021)), SPEDER also shows superior performance, which demonstrates the proposed SPEDER is able to overcome the aforementioned drawback of vanilla spectral representations.

### 6.2 EXPLORATION IN SPARSE-REWARD DEEPMIND CONTROL SUITE

To evaluate the exploration performance of SPEDER, we additionally run experiments on the DeepMind Control Suite. We compare the proposed method with SAC, (including a 2-layer, 3-layer and 5-layer MLP for critic network), PPO, Dreamer-v2 (Hafner et al., 2020), Deep SF (Kulkarni et al., 2016b) and Proto-RL (Yarats et al., 2021). Since the original Dreamer and Proto-RL are designed for image-based control tasks, we adapt them to run the state-based tasks and details can be found at Appendix. F. We run all the algorithms for 200K environment steps across four random seeds

Table 2: Performance on various MuJoCo control tasks. All the results are averaged across 4 random seeds and a window size of 20K. Results marked with $*$ is adopted from MBBL (Wang et al., 2019). SPEDER achieves strong performance compared with baselines.

| | | HalfCheetah | Reacher | Humanoid-ET | Pendulum | I-Pendulum |
|---|---|---|---|---|---|---|
| Model-Based RL | ME-TRPO$^*$ | 2283.7±900.4 | -13.4±5.2 | 72.9±8.9 | **177.3±1.9** | -126.2±86.6 |
| | PETS-RS$^*$ | 966.9±471.6 | -40.1±6.9 | 109.6±102.6 | 167.9±35.8 | -12.1±25.1 |
| | PETS-CEM$^*$ | 2795.3±879.9 | -12.3±5.2 | 110.8±90.1 | 167.4±53.0 | -20.5±28.9 |
| | Best MBBL | 3639.0±1135.8 | **-4.1±0.1** | 1377.0±150.4 | **177.3±1.9** | **0.0±0.0** |
| Model-Free RL | PPO$^*$ | 17.2±84.4 | -17.2±0.9 | 451.4±39.1 | 163.4±8.0 | -40.8±21.0 |
| | TRPO$^*$ | -12.0±85.5 | -10.1±0.6 | 289.8±5.2 | 166.7±7.3 | -27.6±15.8 |
| | SAC$^*$ (3-layer) | 4000.7±202.1 | -6.4±0.5 | **1794.4±458.3** | 168.2±9.5 | -0.2±0.1 |
| Representation RL | DeepSF | 4180.4±113.8 | -16.8±3.6 | 168.6±5.1 | 168.6±5.1 | -0.2±0.3 |
| | SPEDE | 4210.3±92.6 | -7.2±1.1 | 886.9±95.2 | 169.5±0.6 | 0.0±0.0 |
| | **SPEDER** | **5407.9±813.0** | -5.90±0.3 | 1774.875±129.1 | 167.4±3.4 | **0.0±0.0** |
| | | Ant-ET | Hopper-ET | S-Humanoid-ET | CartPole | Walker-ET |
| Model-Based RL | ME-TRPO$^*$ | 42.6±21.1 | 1272.5±500.9 | -154.9±534.3 | 160.1±69.1 | -1609.3±657.5 |
| | PETS-RS$^*$ | 130.0±148.1 | 205.8±36.5 | 320.7±182.2 | 195.0±28.0 | 312.5±493.4 |
| | PETS-CEM$^*$ | 81.6±145.8 | 129.3±36.0 | 355.1±157.1 | 195.5±3.0 | 260.2±536.9 |
| | Best MBBL | 275.4±309.1 | 1272.5±500.9 | **1084.3±77.0** | 200.0±0.0 | 312.5±493.4 |
| Model-Free RL | PPO$^*$ | 80.1±17.3 | 758.0±62.0 | 454.3±36.7 | 86.5±7.8 | 306.1±17.2 |
| | TRPO$^*$ | 116.8±47.3 | 237.4±33.5 | 281.3±10.9 | 47.3±15.7 | 229.5±27.1 |
| | SAC$^*$ (3-layer) | 2012.7±571.3 | 1815.5±655.1 | 834.6±313.1 | 199.4±0.4 | 2216.4±678.7 |
| Representation RL | DeepSF | 768.1±44.1 | 548.9±253.3 | 533.8±154.9 | 194.5±5.8 | 165.6±127.9 |
| | SPEDE | 806.2±60.2 | 732.2±263.9 | 986.4±154.7 | 138.2±39.5 | 501.6±204.0 |
| | **SPEDER** | **1806.8±1488.0** | **2267.6±554.3** | 944.8±354.3 | **200.2±1.0** | **2451.5±1115.6** |

Table 3: Performance on various DeepMind Suite Control tasks. All the results are averaged across four random seeds and a window size of 20K. Comparing with SAC, our method achieves even better performance on sparse-reward tasks. Results are presented in mean ± standard deviation across different random seeds.

| | | cheetah_run | cheetah_run_sparse | walker_run | walker_run_sparse | humanoid_run | hopper_hop |
|---|---|---|---|---|---|---|---|
| Model-Based RL | Dreamer | 542.0 ± 27.7 | **499.9±73.3** | 337.7±67.2 | 95.4±54.7 | 1.0±0.2 | 46.1±17.3 |
| Model-Free RL | PPO | 227.7±57.9 | 5.4±10.8 | 51.6±1.5 | 0.0±0.0 | 1.1±0.0 | 0.7±0.8 |
| | SAC (2-layer) | 222.2±41.0 | 32.4±27.8 | 183.0±23.4 | 53.5±69.3 | 1.3±0.1 | 0.4±0.5 |
| | SAC (3-layer) | **595.2±96.0** | 419.5±73.3 | 700.9±36.6 | 311.5±361.4 | 1.2±0.1 | 28.6±19.5 |
| | SAC (5-layer) | 566.3±123.5 | 364.1±242.3 | **716.9±35.0** | 276.1±319.3 | 8.2±13.8 | 31.1±31.8 |
| Representation RL | DeepSF | 295.3±43.5 | 0.0±0.0 | 27.9±2.2 | 0.1±0.1 | 0.9±0.1 | 0.3±0.1 |
| | Proto RL | 305.5±37.9 | 0.0±0.0 | 433.5±56.8 | 46.9±34.1 | 0.3±0.6 | 1.0±0.2 |
| | **SPEDER** | **593.7±95.1** | 425.5 ± 42.8 | 690.4±20.5 | **683.2±96.0** | **11.5±5.4** | **119.8±89.6** |

with a window size of 20K. From Table 3, we see that SPEDER achieves superior performance compared to SAC using the 2-layer critic network. Compared to SAC and PPO with deeper critic networks, SPEDER has significant gain in tasks with sparse reward (e.g., `walker-run-sparse` and `hopper-hop`).

## 6.3 IMITATION LEARNING PERFORMANCE ON ANTMAZE NAVIGATION

We additionally experiment with using SPEDER features for downstream imitation learning. We consider the challenging AntMaze navigation domain (shown in Figure 3) from the D4RL (Fu et al., 2020), which consists of a 8-DoF quadraped robot whose task is to navigate towards a goal position in the maze environment. We compare SPEDER to several recent state-of-the-art for pre-training representations from suboptimal offline data, including OPAL (Ajay et al., 2020), SPiRL (Pertsch et al., 2020), SkiLD (Pertsch et al., 2021), and TRAIL (Yang et al., 2021). For OPAL, SPiRL, and SkiLD, we use horizons of $t = 1$ and $t = 10$ for learning temporally-extended skills. For TRAIL, we report the performance of the TRAIL energy-based model (EBM) as well as the TRAIL Linear model with random Fourier features (Rahimi & Recht, 2007).

Following the behavioral cloning setup in (Yang et al., 2021), we use a set of 10 expert trajectories of the agent navigating from one corner of the maze to the opposite corner as the expert dataset $\mathcal{D}^{\pi^*}$. For the suboptimal dataset $\mathcal{D}^{\text{off}}$, we use the "diverse" datasets from D4RL (Fu et al., 2020), which consist of 1M samples of the agent navigating from different initial locations to different goal positions. We report the average return on AntMaze tasks, and observe that SPEDER achieves comparable performance as other state-of-the-art representations on downstream imitation learning in Figure 4. The comparison and experiment details can be found in Appendix F.

## 7 CONCLUSION

We have proposed a novel objective, Spectral Decomposition Representation (SPEDER), that factorizes the state-action transition kernel to obtain policy-independent spectral features. We show how to use the representations obtained with SPEDER to perform sample efficient online and offline RL, as well as imitation learning. We provide a thorough theoretical analysis of SPEDER and empirical comparisons on multiple RL benchmarks, demonstrating the effectiveness of SPEDER.

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

## A    MORE RELATED WORK

Representation learning in RL has attracted more attention in recent years. Within model-based RL (MBRL), many methods for learning representations of the reward and the dynamics have been proposed. Several recent MBRL methods learn latent state representations to be used for planning in latent space as a way to improve model-based policy optimization (Oh et al., 2017; Silver et al., 2018; Racanière et al., 2017; Hafner et al., 2019).

Beyond MBRL, there also exist many algorithms for learning useful state representations to accelerate RL. For example, recent works have introduced unsupervised auxiliary losses to significantly improve RL performance (Pathak et al., 2017; Oord et al., 2018; Laskin et al., 2020; Jaderberg et al., 2016). Contrastive losses  (Oord et al., 2018; Anand et al., 2019; Srinivas et al., 2020; Stooke et al., 2021), which encourage similar states to be closer in embedding space, where the notion of similarity is usually defined in terms of temporal distance (Anand et al., 2019; Sermanet et al., 2018) or image-based data augmentations (Srinivas et al., 2020), also show promising performance. Within goal-conditioned RL (Kaelbling, 1993; Schaul et al., 2015; Andrychowicz et al., 2017), various representation learning algorithms have been proposed to handle high-dimensional observation and goal spaces, such as using a variational autoencoder (Nair et al., 2018; Pong et al., 2019), or representations that explicitly capture useful information for control, while ignoring irrelevant factors of variation in the observation (Ghosh et al., 2018; Lee et al., 2020).

Beyond these representations on the state space, there are other kinds of representations that are designed for specific tasks. For example, Touati & Ollivier (2021) proposed to deal with the reward transfer task by learning a reward-dependent feature $F(s, a, r)$ such that the greedy policy with respect to $F(s, a, r)^\top r$ is optimal under $r$.

## B    IMPLEMENTATION DETAILS

In this section, we provide more implementation details of SPEDER for online exploration.

- **Representation Learning.** We parameterize the representation network $\phi_\theta(s, a)$ and $\mu(s')$, and optimize the representation in Line 6 in Algorithm 1 with the data collected in the replay buffer $\mathcal{D}$, via minimizing the following objective:

$$
\mathcal{L}(\phi, \mu) := -\frac{1}{|\mathcal{D}|} \sum_{(s_i, a_i, s_{i+1}) \in \mathcal{D}} \left[\phi(s_i, a_i)^\top \mu(s_{i+1}) p(s_{i+1})\right] + \frac{1}{2d|\mathcal{D}_{\text{base}}|} \sum_{s_j \in \mathcal{D}_{\text{base}}} \left[p(s_j) \mu(s_j)^\top \mu(s_j)\right]
$$

$$
+ \frac{\lambda_{\text{ortho}}}{|\mathcal{D}|^2} \sum_{(s_i, a_i) \sim \mathcal{D}} \sum_{(s'_i, a'_i) \sim \mathcal{D}} \left[\sum_{j,k \in [d]} \left(\phi_j(s_i, a_i) \phi_k(s_i, a_i) - \frac{\delta_{jk}}{d}\right) \left(\phi_j(s'_i, a'_i) \phi_k(s'_i, a'_i) - \frac{\delta_{jk}}{d}\right)\right]
$$

$$
+ \frac{\lambda_{\text{prob}}}{|\mathcal{D}|} \sum_{(s_i, a_i) \in \mathcal{D}} \left[\left(\log \frac{1}{|\mathcal{D}_{\text{base}}|} \sum_{s_j \in \mathcal{D}_{\text{base}}} \phi(s_i, a_i)^\top \mu(s_j)\right)^2\right],
$$

where $p(s)$ is a base measure on the state space and $|\mathcal{D}_{\text{base}}| = \{s_j\}$ where $s_j \sim p(s)$, $\lambda_{\text{ortho}}$ and $\lambda_{\text{prob}}$ are coefficients of the regularizers that can help enforce $\phi$ to be orthogonal (see Wu et al. (2018) for more details) and $\phi(s, a)^\top \mu(s')$ to be a valid conditional density (see Ma & Collins (2018) for more details) accordingly.

- **Planning module.** We implement Line 9 in Algorithm 1 with SAC algorithm (Haarnoja et al., 2018) upon the learned feature. Specifically,

  - We parameterize the critic network $Q_\theta$ as a two-layer MLP on top of the representation $\phi_\theta(s, a)$, whose parameter will be frozen.
  - The critic network and the actor network in SAC are both updated with the samples collected in the replay buffer.

- **Exploration bonus.** We can optionally add the exploration bonus (Line 8 in Algorithm 1) as we discussed in the main text.

## C  ALGORITHM FOR OFFLINE POLICY OPTIMIZATION

For completeness, we include the algorithm for offline policy optimization with SPEDER here.

---

**Algorithm 2** Offline Policy Optimization with SPEDER

---

1: **Input:** Regularizer $\lambda$, Parameter $\alpha$, Model class $\mathcal{F}$, Dataset $\mathcal{D}$ sampled from the stationary distribution of the behavior policy $\pi_b$.
2: Learn representation $\widehat{\phi}(s,a)$ with $\mathcal{D}$ via equation 10.
3: Set the empirical covariance matrix

$$\widehat{\Sigma} = \sum_{(s,a)\in\mathcal{D}} \widehat{\phi}(s,a)\widehat{\phi}(s,a)^\top + \lambda I.$$

4: Set the reward penalty: $\widehat{b}(s,a) = \alpha\sqrt{\widehat{\phi}(s,a)^\top\widehat{\Sigma}^{-1}\widehat{\phi}(s,a)}$.
5: Solve $\widehat{\pi} = \arg\max_\pi V^\pi_{\widehat{P}, r-\widehat{b}}$.
6: **Return** $\widehat{\pi}$

---

## D  PROOF DETAILS

### D.1  NON-ASYMPTOTIC GENERALIZATION BOUND

In this subsection, we consider the non-asymptotic generalization bound for the $\ell_2$ minimization, which is necessary for the proof of series of key lemmas (Lemma 7 and Lemma 15) that are used in the PAC guarantee of the online and offline reinforcement learning. For simplicity, we denote the instance space as $\mathcal{X}$ and the target space as $\mathcal{Y}$, and we want to estimate the conditional density $p(y|x) = f^*(x, y)$. Assume we are given a function class $\mathcal{F} : (\mathcal{X} \times \mathcal{Y}) \to \mathbb{R}$ with $f^* \in \mathcal{F}$, as well as the data $\mathcal{D} := \{(x_i, y_i)\}_{i=1}^n$, where $x_i \sim \mathcal{D}_i(x_{1:i-1}, y_{1:i-1})$, $y_i \sim p(\cdot|x_i)$ and $\mathcal{D}_i$ is some data generating process that depends on the previous samples (a.k.a a martingale process). We define the tangent sequence $\mathcal{D}' := \{(x_i', y_i')\}$ where $x_i' \sim \mathcal{D}_i(x_{1:i-1}, y_{1:i-1})$ and $y_i' \sim p(\cdot|x_i')$. Consider the estimator obtained by following minimization problem:

$$\widehat{f} = \arg\min_{f\in\mathcal{F}} \left\{ \sum_{i=1}^n -2f(x_i, y_i) + \sum_{i=1}^n \left( \sum_{y\in\mathcal{Y}} f^2(x_i, y) \right) \right\}, \tag{14}$$

where the summation over the counting measure of $\mathcal{Y}$ for discrete case can be replaced by the integration over the Lebesgue measure of $\mathcal{Y}$ for continuous case. We first prove the following decoupling inequality motivated by Lemma 24 of (Agarwal et al., 2020).

**Lemma 4.** *Let $L(f, \mathcal{D}) = \sum_{i=1}^n \ell(f, (x_i, y_i))$, $\mathcal{D}'$ is a tangent sequence of $\mathcal{D}$ and $\widehat{f}(\mathcal{D})$ be any estimator taking random variable $\mathcal{D}$ as input with range $\mathcal{F}$. Then*

$$\mathbb{E}_{\mathcal{D}} \left[ \exp\left( -L(\widehat{f}(\mathcal{D}), \mathcal{D}) - \log\mathbb{E}_{\mathcal{D}'}[\exp[-L(\widehat{f}(\mathcal{D}), \mathcal{D}')]] - \log|\mathcal{F}| \right) \right] \leqslant 1.$$

*Proof.* Let $\pi$ be the uniform distribution over $\mathcal{F}$ and $g : \mathcal{F} \to \mathbb{R}$ be any function. Define the following probability measure over $\mathcal{F}$: $\mu(f) = \frac{\exp g(f)}{\sum_{f\in\mathcal{F}} \exp(g(f))}$. Then for any probability distribution $\widehat{\pi}$ over $\mathcal{F}$, we have:

$$0 \leqslant \text{KL}(\widehat{\pi}\|\mu)$$
$$= \sum_{f\in\mathcal{F}} \widehat{\pi}(f) \log\frac{\widehat{\pi}(f)}{\mu(f)}$$
$$= \sum_{f\in\mathcal{F}} [\widehat{\pi}(f)\log\widehat{\pi}(f) - \widehat{\pi}(f)g(f)] + \log\sum_{f\in\mathcal{F}} \exp(g(f))$$

$$= \sum_{f \in \mathcal{F}} [\widehat{\pi}(f) \log \widehat{\pi}(f) + \widehat{\pi}(f) \log |\mathcal{F}|] - \sum_{f \in \mathcal{F}} \widehat{\pi}(f) g(f) + \log \mathbb{E}_{f \sim \pi} \exp(g(f))$$

$$= \mathrm{KL}(\widehat{\pi} \| \pi) - \sum_{f \in \mathcal{F}} \widehat{\pi}(f) g(f) + \log \mathbb{E}_{f \sim \pi} \exp(g(f))$$

$$\leqslant \log |\mathcal{F}| - \sum_{f \in \mathcal{F}} \widehat{\pi}(f) g(f) + \log \mathbb{E}_{f \sim \pi} \exp(g(f)).$$

Re-arranging, we have that

$$\sum_f \widehat{\pi}(f) g(f) - \log |\mathcal{F}| \leqslant \log \mathbb{E}_{f \sim \pi} \exp(g(f)).$$

Take $g = -L(f, \mathcal{D}) - \log \mathbb{E}_{\mathcal{D}'}[\exp(-L(f, \mathcal{D}'))], \widehat{\pi}(f) = \mathbf{1}_{\widehat{f}(\mathcal{D})}$, we obtain that for any $\mathcal{D}$,

$$-L(\widehat{f}(\mathcal{D}), \mathcal{D}) - \log \mathbb{E}_{\mathcal{D}'} \exp(-L(\widehat{f}(\mathcal{D}), \mathcal{D}')) - \log |\mathcal{F}| \leqslant \log \mathbb{E}_{f \sim \pi} \frac{\exp(-L(\widehat{f}(\mathcal{D}), \mathcal{D}))}{\mathbb{E}_{\mathcal{D}'} \exp(-L(\widehat{f}(\mathcal{D}), \mathcal{D}'))}.$$

We exponentiate both sides and take the expectation over $\mathcal{D}$, which gives

$$\mathbb{E}_{\mathcal{D}} \left[ \exp \left( -L(\widehat{f}(\mathcal{D}), \mathcal{D}) - \log \mathbb{E}_{\mathcal{D}'} \left[ \exp(-L(\widehat{f}(\mathcal{D}), \mathcal{D}')) \right] - \log |\mathcal{F}| \right) \right] \leqslant \mathbb{E}_{f \sim \pi} \mathbb{E}_{\mathcal{D}} \frac{\exp(-L(\widehat{f}(\mathcal{D}), \mathcal{D}))}{\mathbb{E}_{\mathcal{D}'} \exp \left[ -L(\widehat{f}(\mathcal{D}), \mathcal{D}') | \mathcal{D} \right]}.$$

Note that, conditioned on $\mathcal{D}$, the samples in the tangent sequence $\mathcal{D}'$ are independent, which leads to

$$\mathbb{E}_{\mathcal{D}'} \exp \left[ -L(\widehat{f}(\mathcal{D}), \mathcal{D}') | \mathcal{D} \right] = \prod_{i=1}^{n} \exp \left( \mathbb{E}_{(x_i, y_i) \sim \mathcal{D}_i} [-l(f, (x_i, y_i))] \right).$$

As a result, we can peel off terms from $n$ down to 1 and cancel out terms in the numerator. Hence, we have

$$\mathbb{E}_{\mathcal{D}} \left[ -\exp \left( L(\widehat{f}(\mathcal{D}), \mathcal{D}) - \log \mathbb{E}_{\mathcal{D}'} \left[ \exp \left( -L(\widehat{f}(\mathcal{D}), \mathcal{D}') \right) \right] - \log |\mathcal{F}| \right) \right] \leqslant 1,$$

which concludes the proof. $\qquad \square$

**Theorem 5.** *Assume $|\mathcal{F}| < \infty$, $f^* \in \mathcal{F}$ and $\|f(x, y)\|_\infty \leqslant C, \forall f \in \mathcal{F}$. Then with probability at least $1 - \delta$, we have*

$$\sum_{i=1}^{n} \mathbb{E}_{x_i \sim \mathcal{D}_i} \|f^* - f\|_2^2 \leqslant C' \log |\mathcal{F}|/\delta,$$

*where $C'$ only depends on $C$.*

With Chernoff's method, we have that

$$- \log \mathbb{E}_{\mathcal{D}'} \left[ \exp \left( -L(\widehat{f}(\mathcal{D}), \mathcal{D}') \right) \right] \leqslant L(\widehat{f}(\mathcal{D}), \mathcal{D}) + \log |\mathcal{F}| + \log 1/\delta.$$

Take

$$l(f, (x_i, y_i)) = 2(f^*(x_i, y_i) - f(x_i, y_i)) + \sum_{y \in \mathcal{Y}} \left( f(x_i, y)^2 - f^*(x_i, y)^2 \right),$$

and

$$L(f, \mathcal{D}) = \rho \left( \sum_{i=1}^{n} 2(f^*(x_i, y_i) - f(x_i, y_i)) + \sum_{i=1}^{n} \sum_{y \in \mathcal{Y}} \left( f(x_i, y)^2 - f^*(x_i, y)^2 \right) \right),$$

where $\rho > 0$ is a constant to determine later. As $\widehat{f}(\mathcal{D})$ is obtained by minimizing $L(f, \mathcal{D})$, and $f^* \in \mathcal{F}$, we have $L(\widehat{f}(\mathcal{D}), \mathcal{D}) \leqslant L(f^*, \mathcal{D}) \leqslant 0$. Furthermore, as $\mathcal{D}'$ is the tangent sequence of $\mathcal{D}$, direct computation shows

$$- \log \mathbb{E}_{\mathcal{D}'} \left[ \exp \left( -L(\widehat{f}(\mathcal{D}), \mathcal{D}') \right) \right] \leqslant \log \frac{|\mathcal{F}|}{\delta}.$$

We now relate the term $-\log \mathbb{E}_{\mathcal{D}'}\left[\exp\left(-L(\widehat{f}(\mathcal{D}), \mathcal{D}')\right)\right]$ with our target $\sum_{i=1}^n \mathbb{E}_{x_i \sim \mathcal{D}_i}\|\widehat{f}(x_i, \cdot) - f^*(x_i, \cdot)\|_2^2$ using the method introduced in Zhang (2006).

Note that $\sum_{y \in \mathcal{Y}} f(x, y) = 1$, as $\|f\|_\infty \leqslant C$, with a straightforward application of Hölder's inequality, we have that $\sum_{y \in \mathcal{Y}} f(x, y)^2 \leqslant C$. We then consider the term

$$\mathbb{E}_{y_i \sim f^*(x_i, y)}\left[l(f, (x_i, y_i))^2\right] + \mathbb{E}_{y_i \sim f(x_i, y)}\left[l(f, (x_i, y_i))^2\right]$$

$$=4 \sum_{y \in \mathcal{Y}}\left[(f(x_i, y) + f^*(x_i, y))(f(x_i, y) - f^*(x_i, y))^2\right] - 3\left(\sum_{y \in \mathcal{Y}}(f^*(x_i, y)^2 - f(x_i, y)^2)\right)^2$$

$$\leqslant \sum_{y \in \mathcal{Y}}\left((f(x_i, y) - f^*(x_i, y))^2\right)\left(8C + 3\sum_{y \in \mathcal{Y}}(f(x_i, y) + f^*(x_i, y))^2\right)$$

$$\leqslant 20C \sum_{y \in \mathcal{Y}}\left((f(x_i, y) - f^*(x_i, y))^2\right)$$

$$=20C\mathbb{E}_{y_i \sim f^*(x, y)}\left[l(f, (x_i, y_i))\right].$$

As $\mathbb{E}_{y_i \sim f(x_i)}\left[l(f, (x_i, y_i))^2\right] \geqslant 0$, we can conclude that

$$\mathbb{E}_{y_i \sim f^*(x_i, y)}\left[l(f, (x_i, y_i))^2\right] \leqslant \left(20C\mathbb{E}_{y_i \sim f^*(x, y)}\left[l(f, (x_i, y_i))\right]\right).$$

Furthermore, it is straightforward to see $|l(f, (x_i, y_i))| \leqslant 3C$. With the last bound in Proposition 1.2 in Zhang (2006), we have that

$$\log \mathbb{E}_{\mathcal{D}'}\left[\exp\left(-L(\widehat{f}(\mathcal{D}), \mathcal{D}')\right)\right]$$

$$=\sum_{i=1}^n \log \mathbb{E}_{(x_i, y_i) \sim \mathcal{D}_i}\left[\exp(-\rho l(f, (x_i, y_i)))\right]$$

$$\leqslant -\rho \sum_{i=1}^n \mathbb{E}_{(x_i, y_i) \sim \mathcal{D}_i}\left[l(f, (x_i, y_i))\right] + \frac{\exp(3\rho C) - 3\rho C - 1}{9C^2}\mathbb{E}_{(x_i, y_i) \sim \mathcal{D}_i}\left[l(f, (x_i, y_i))^2\right]$$

$$\leqslant -\left(\rho - \frac{20(\exp(3\rho C) - 3\rho C - 1)}{9C}\right)\sum_{i=1}^n \mathbb{E}_{(x_i, y_i) \sim \mathcal{D}_i}\|\widehat{f}(x_i, \cdot) - f^*(x_i, \cdot)\|_2^2.$$

As $\exp(x) - x - 1 \approx 0.5x^2$ as $x \to 0$, we know there exists sufficiently small $\rho$ that only depends on $C$, such that $9\rho C > 20(\exp(3\rho C) - 3\rho C - 1)$. Hence, we know that,

$$\mathbb{E}_{(x_i, y_i) \sim \mathcal{D}_i}\|\widehat{f}(x_i, \cdot) - f^*(x_i, \cdot)\|_2^2 \leqslant \frac{9C}{9\rho C - 20(\exp(3\rho C) - 3\rho C - 1)}\log\frac{|\mathcal{F}|}{\delta}.$$

**Compared with the MLE guarantee** For discrete domain, as $L_2$ norm is always bounded by $L_1$ norm, our guarantee is weaker than the guarantee of MLE used in (Agarwal et al., 2020; Uehara et al., 2022). However, for general cases, $L_1$ and $L_2$ does not imply each other, and hence we cannot directly compare our theoretical guarantee with the MLE guarantee. Nevertheless, our method is easier to optimize compared to the MLE, which makes it a preferable practical choice.

### D.2 PAC BOUNDS FOR ONLINE REINFORCEMENT LEARNING

Before we start, we first state some basic properties of MDP that can be obtained from the definition of the related terms. For the state visitation distribution, a straightforward computation shows that

$$d_P^\pi(s) = (1 - \gamma)\rho(s) + \gamma\mathbb{E}_{\tilde{s} \sim d_P^\pi, \tilde{a} \sim \pi(\cdot|\tilde{s})}P(s|\tilde{s}, \tilde{a}).$$

Meanwhile, we have that

$$V_{P,r}^\pi = \frac{1}{1 - \gamma}\mathbb{E}_{s \sim d_P^\pi, a \sim \pi(\cdot|s)}[r(s, a)].$$

For now, we assume $\widehat{P}_n(\cdot, \cdot)$ is a valid probability measure, $P \in \mathcal{F}$, and the following two inequalities hold $\forall n \in \mathbb{N}^+$ with probability at least $1 - \delta$:

$$\mathbb{E}_{s \sim \rho_n, a \sim \mathcal{U}(\mathcal{A})} \|\widehat{P}_n(\cdot|s, a) - P(\cdot|s, a)\|_2^2 \leqslant \zeta_n$$

$$\mathbb{E}_{s \sim \rho_n', a \sim \mathcal{U}(\mathcal{A})} \|\widehat{P}_n(\cdot|s, a) - P(\cdot|s, a)\|_2^2 \leqslant \zeta_n$$

**Proof Sketch**  Our proof is organized as follows:

- Based on Theorem 5, we prove a one-step back inequality for the learned model (Lemma 7), which is further used to show the optimisticity, *i.e.*, the policy planning on the learned model with the additional bonus upper bound the optimal value up to some error term (Lemma 9).

- We then bound the cumulative regret of the adaptive chosen policy (Lemma 13) based on the established optimisticity, and further exploit a one-step back inequality for the true model (Lemma 10) and standard elliptical potential lemma (Lemma 20).

- With standard regret to PAC conversion, we obtain the final PAC guarantee (Theorem 14).

We first state the following basic property for the value function:

**Lemma 6** ($L_2$ norm of $V_{P,r}^\pi$). *For any policy $\pi$, we have that*

$$\|V_{P,r}^\pi\|_2 \leqslant \sqrt{2d \left(1 + \frac{d\gamma^2}{(1-\gamma)^2}\right)} \lesssim \frac{d}{1-\gamma}$$

*Proof.* From the properties of low-rank MDP, we know there exists $w^\pi$, $\|w^\pi\|_2 \leqslant \frac{\sqrt{d}}{1-\gamma}$ and $Q_{P,r}^\pi(s, a) = \phi^*(s, a)^\top w_h^\pi$. Then we have

$$\|V_{P,r}^\pi\|_2^2 = \int_{\mathcal{S}} V^\pi(s)^2 \, ds$$

$$= \int_{\mathcal{S}} \left(\int_{\mathcal{A}} \pi(a|s)(r(s, a) + \gamma Q_{P,r}^\pi(s, a)) \, da\right)^2 ds$$

$$\leqslant \int_{\mathcal{S}} \left(\int_{\mathcal{A}} r(s, a) + \gamma Q_{P,r}^\pi(s, a) \, da\right)^2 ds$$

$$\leqslant 2 \int_{\mathcal{S}} \left[\int_{\mathcal{A}} r(s, a) \, da\right]^2 ds + 2\gamma^2 \int_{\mathcal{S}} \left[\int_{\mathcal{A}} Q_{P,r}^\pi(s, a) \, da\right]^2 ds$$

$$\leqslant 2d + \frac{2d\gamma^2}{(1-\gamma)^2} \int_{\mathcal{S}} \left[\int_{\mathcal{A}} \|\phi^*(s, a)\|_2 \, da\right]^2 ds$$

$$\leqslant 2d \left(1 + \frac{d\gamma^2}{(1-\gamma)^2}\right)$$

$$\lesssim \frac{d^2}{(1-\gamma)^2},$$

which concludes the proof. $\qquad\square$

Before we proceed to the proof, we first define the following terms. Let $\rho_n(s) = \frac{1}{n} \sum_{i=1}^n d_{P^*}^{\pi_i}(s)$. With slightly abuse of notation, we also use $\rho_n(s, a) = \frac{1}{n} \sum_{i=1}^n d_{P^*}^{\pi_i}(s, a)$, and use $\rho_n'$ to denote the marginal distribution of $s'$ for the triple $(s, a, s') \sim \rho_n(s)\mathcal{U}(a)P^*(s'|s, a)$. For notation simplicity, we denote

$$\Sigma_{\rho_n \times \mathcal{U}(\mathcal{A}), \phi} = n\mathbb{E}_{s \sim \rho_n, a \sim \mathcal{U}(\mathcal{A})} \left[\phi(s, a)\phi(s, a)^\top\right] + \lambda_n I,$$

$$\Sigma_{\rho_n, \phi} = n\mathbb{E}_{(s,a) \sim \rho_n} \left[\phi(s, a)\phi(s, a)^\top\right] + \lambda_n I,$$

$$\widehat{\Sigma}_{n, \phi} = n\mathbb{E}_{(s,a) \sim \mathcal{D}_n} \left[\phi(s, a)\phi(s, a)^\top\right] + \lambda_n I.$$

The following lemmas will be helpful when we demonstrate the effectiveness of our bonus:

**Lemma 7** (One-step back inequality for the learned model). *Assume $g : \mathcal{S} \times \mathcal{A} \to \mathbb{R}$ satisfies that* $\|g\|_\infty \leqslant B_\infty$, $\left\|\int_\mathcal{A} g(\cdot, a)\,\mathrm{d}a\right\|_2 \leqslant B_2$, *then we have that*

$$\left|\mathbb{E}_{(s,a)\sim d^\pi_{\widehat{P}_n}}\{g(s,a)\}\right| \leqslant \sqrt{(1-\gamma)|\mathcal{A}|\mathbb{E}_{s\sim\rho_n,a\sim\mathcal{U}(\mathcal{A})}\{g^2(s,a)\}}$$

$$+ \gamma\sqrt{n|\mathcal{A}|\mathbb{E}_{s\sim\rho'_n,a\sim\mathcal{U}(\mathcal{A})}\{g^2(s,a)\} + B_2^2 n\zeta_n + \lambda_n B_\infty^2 d} \cdot \mathbb{E}_{(\tilde{s},\tilde{a})\sim d^\pi_{\widehat{P}_n}}\left[\left\|\widehat{\phi}_n(\tilde{s},\tilde{a})\right\|_{\Sigma^{-1}_{\rho_n\times\mathcal{U},\widehat{\phi}_n}}\right].$$

*Proof.* Note that

$$\mathbb{E}_{(s,a)\sim d^\pi_{\widehat{P}_n}}\{g(s,a)\} = \gamma\mathbb{E}_{(\tilde{s},\tilde{a})\sim d^\pi_{\widehat{P}_n},s\sim\widehat{P}_n(\cdot|\tilde{s},\tilde{a}),a\sim\pi(\cdot|s)}\{g(s,a)\} + (1-\gamma)\mathbb{E}_{s\sim\rho,a\sim\pi(\cdot|s)}\{g(s,a)\}.$$

For the second term, note that $d^\pi_P(s) \geqslant (1-\gamma)\rho(s)$, hence

$$(1-\gamma)\mathbb{E}_{s\sim\rho,a\sim\pi(\cdot|s)}\{g(s,a)\}$$

$$\leqslant (1-\gamma)\sqrt{\mathbb{E}_{s\sim\rho,a\sim\pi(\cdot|s)}\{g^2(s,a)\}}$$

$$= (1-\gamma)\sqrt{\mathbb{E}_{s\sim\rho_n,a\sim\mathcal{U}(\mathcal{A})}\left\{\frac{\rho(s)\pi(a|s)|\mathcal{A}|}{\rho_n(s)}g^2(s,a)\right\}}$$

$$\leqslant \sqrt{(1-\gamma)|\mathcal{A}|\mathbb{E}_{s\sim\rho_n,a\sim\mathcal{U}(\mathcal{A})}\{g^2(s,a)\}}.$$

For the first term, we have that

$$\mathbb{E}_{(\tilde{s},\tilde{a})\sim d^\pi_{\widehat{P}_n},s\sim\widehat{P}_n(\cdot|\tilde{s},\tilde{a}),a\sim\pi(\cdot|s)}\{g(s,a)\}$$

$$= \mathbb{E}_{(\tilde{s},\tilde{a})\sim d^\pi_{\widehat{P}_n}}\widehat{\phi}_n(\tilde{s},\tilde{a})^\top\left[\int_{\mathcal{S}\times\mathcal{A}}\widehat{\mu}_n(s)\pi(a|s)g(s,a)\,\mathrm{d}s\,\mathrm{d}a\right]$$

$$\leqslant \mathbb{E}_{(\tilde{s},\tilde{a})\sim d^\pi_{\widehat{P}_n}}\left\|\widehat{\phi}_n(\tilde{s},\tilde{a})\right\|_{\Sigma^{-1}_{\rho_n\times\mathcal{U},\widehat{\phi}_n}}\left\|\int_{\mathcal{S}\times\mathcal{A}}\widehat{\mu}_n(s)\pi(a|s)g(s,a)\,\mathrm{d}s\,\mathrm{d}a\right\|_{\Sigma_{\rho_n\times\mathcal{U},\widehat{\phi}_n}},$$

where for the inequality we use the generalized Cauchy-Schwartz inequality. Note

$$\left\|\int_{\mathcal{S}\times\mathcal{A}}\widehat{\mu}_n(s)\pi(a|s)g(s,a)\,\mathrm{d}s\,\mathrm{d}a\right\|^2_{\Sigma_{\rho_n\times\mathcal{U},\widehat{\phi}_n}}$$

$$= n\mathbb{E}_{\tilde{s}\sim\rho_n,\tilde{a}\sim\mathcal{U}(\mathcal{A})}\left[\left(\int_{\mathcal{S}\times\mathcal{A}}\widehat{P}_n(s|\tilde{s},\tilde{a})\pi(a|s)g(s,a)\,\mathrm{d}s\,\mathrm{d}a\right)^2\right] + \lambda_n\left\|\int_{\mathcal{S}\times\mathcal{A}}\widehat{\mu}_n(s)\pi(a|s)g(s,a)\,\mathrm{d}s\,\mathrm{d}a\right\|^2$$

$$\leqslant 2n\mathbb{E}_{\tilde{s}\sim\rho_n,\tilde{a}\sim\mathcal{U}(\mathcal{A})}\left[\left(\int_{\mathcal{S}\times\mathcal{A}}P(s|\tilde{s},\tilde{a})\pi(a|s)g(s,a)\,\mathrm{d}s\,\mathrm{d}a\right)^2\right]$$

$$+ 2n\mathbb{E}_{\tilde{s}\sim\rho_n,\tilde{a}\sim\mathcal{U}(\mathcal{A})}\left[\left(\int_{\mathcal{S}\times\mathcal{A}}(\widehat{P}_n(s|\tilde{s},\tilde{a}) - P(s|\tilde{s},\tilde{a}))\pi(a|s)g(s,a)\,\mathrm{d}s\,\mathrm{d}a\right)^2\right] + \lambda_n B_\infty^2 d.$$

With Jensen's inequality, we have

$$\mathbb{E}_{\tilde{s}\sim\rho_n,\tilde{a}\sim\mathcal{U}(\mathcal{A})}\left[\left(\int_{\mathcal{S}\times\mathcal{A}}P(s|\tilde{s},\tilde{a})\pi(a|s)g(s,a)\,\mathrm{d}s\,\mathrm{d}a\right)^2\right]$$

$$\leqslant \mathbb{E}_{\tilde{s}\sim\rho_n,\tilde{a}\sim\mathcal{U}(\mathcal{A}),s\sim P(\cdot|\tilde{s},\tilde{a}),a\sim\pi(\cdot|s)}\{g^2(s,a)\}$$

$$= \mathbb{E}_{s\sim\rho'_n,a\sim\pi(\cdot|s)}\{g^2(s,a)\}$$

$$\leqslant |\mathcal{A}|\mathbb{E}_{s\sim\rho'_n,a\sim\mathcal{U}(\mathcal{A})}\{g^2(s,a)\}$$

Meanwhile,

$$\mathbb{E}_{\tilde{s}\sim\rho_n,\tilde{a}\sim\mathcal{U}(\mathcal{A})}\left[\left(\int_{\mathcal{S}\times\mathcal{A}}(\widehat{P}_n(s|\tilde{s},\tilde{a}) - P(s|\tilde{s},\tilde{a}))\pi(a|s)g(s,a)\,\mathrm{d}s\,\mathrm{d}a\right)^2\right]$$

$$\leqslant \mathbb{E}_{\tilde{s}\sim\rho_n,\tilde{a}\sim\mathcal{U}(\mathcal{A})}\left[\left\|\widehat{P}_n(\cdot|\tilde{s},\tilde{a})-P(\cdot|\tilde{s},\tilde{a})\right\|_2^2\left\|\int_{\mathcal{A}}\pi(a|\cdot)g(\cdot,a)\,\mathrm{d}a\right\|_2^2\right]$$

$$\leqslant \mathbb{E}_{\tilde{s}\sim\rho_n,\tilde{a}\sim\mathcal{U}(\mathcal{A})}\left[\left\|\widehat{P}_n(\cdot|\tilde{s},\tilde{a})-P(\cdot|\tilde{s},\tilde{a})\right\|_2^2\left\|\int_{\mathcal{A}}g(\cdot,a)\,\mathrm{d}a\right\|_2^2\right]$$

$$\leqslant B_2^2\zeta_n,$$

where the last inequality is due to Theorem 5. Substitute back, we obtain the desired result. $\qquad\square$

**Lemma 8** (Concentration of the bonus term). *Let $\lambda_n = \Theta(d\log(n|\mathcal{F}|/\delta))$, and define:*

$$\Sigma_{\rho_n\times\mathcal{U},\phi} = n\mathbb{E}_{s\sim\rho_n,a\sim\mathcal{U}(\mathcal{A})}\phi(s,a)\phi(s,a)^\top + \lambda_n I,$$

$$\widehat{\Sigma}_{n,\phi} = \sum_{i\in[n]}\phi(s_i,a_i)\phi(s_i,a_i)^\top + \lambda_n I.$$

*Then there exists absolute constant $c_1$ and $c_2$, such that*

$$\forall n\in\mathbb{N}^+, \forall\phi\in\Phi, c_1\|\phi(s,a)\|_{\Sigma_{\rho_n\times\mathcal{U},\phi}^{-1}} \leqslant \|\phi(s,a)\|_{\widehat{\Sigma}_{n,\phi}^{-1}} \leqslant c_2\|\phi(s,a)\|_{\Sigma_{\rho_n\times\mathcal{U},\phi}^{-1}}$$

*which holds with probability at least $1-\delta$.*

*Proof.* See (Uehara et al., 2022, Lemma 11). $\qquad\square$

With these lemmas, we are now ready to show the optimism.

**Lemma 9** (Optimism). *Let*

$$\alpha_n = \Theta\left(\frac{d\sqrt{|\mathcal{A}|n\zeta_n}}{1-\gamma}\right),$$

$$\lambda_n = \Theta\left(d\log(n|\mathcal{F}|/\delta)\right),$$

*then for any policy $\pi$ we have*

$$V_{\widehat{P}_n,r+b_n}^\pi \geqslant V_{P,r}^\pi - \sqrt{\frac{2|\mathcal{A}|d\left(1+\frac{\gamma^2 d}{(1-\gamma)^2}\right)\zeta_n}{(1-\gamma)}}.$$

*Proof.* With the simulation lemma (*i.e.*, Lemma 19), we have that

$$V_{\widehat{P}_n,r+b_n}^\pi - V_{P,r}^\pi$$
$$= \frac{1}{1-\gamma}\mathbb{E}_{(s,a)\sim d_{\widehat{P}_n}^\pi}\left[b_n(s,a)+\gamma\mathbb{E}_{\widehat{P}_n(s'|s,a)}\left[V_{P,r}^\pi(s')\right]-\gamma\mathbb{E}_{P(s'|s,a)}\left[V_{P,r}^\pi(s')\right]\right].$$

Consider the function $g$ on $\mathcal{S}\times\mathcal{A}$ defined as follows:

$$g(s,a) := \left|\mathbb{E}_{P(s'|s,a)}\left[V_{P,r}^\pi(s')\right]-\mathbb{E}_{\widehat{P}_n(s'|s,a)}\left[V_{P,r}^\pi(s')\right]\right|.$$

With Hölder's inequality, we have that $\|g\|_\infty \leqslant \frac{2}{1-\gamma}$. Furthermore, as

$$g(s,a) \leqslant \phi^*(s,a)^\top\int_{\mathcal{S}}\mu^*(s')V_{P,r}^\pi(s')\,\mathrm{d}s' + \widehat{\phi}_n(s,a)^\top\int_{\mathcal{S}}\widehat{\mu}_n^\top(s')V_{P,r,h+1}^\pi(s')\,\mathrm{d}s'$$

$$\leqslant \|\phi^*(s,a)\|\left\|\int_{\mathcal{S}}\mu^*(s')V_{P,r}^\pi(s')\,\mathrm{d}s'\right\| + \left\|\widehat{\phi}_n(s,a)\right\|\left\|\int_{\mathcal{S}}\widehat{\mu}_n^\top(s')V_{P,r}^\pi(s')\,\mathrm{d}s'\right\|$$

$$\leqslant \frac{\sqrt{d}}{1-\gamma}\left(\|\phi^*(s,a)\|+\left\|\widehat{\phi}_n(s,a)\right\|\right)$$

where the first inequality is due to the triangle inequality; the second inequality is from the Cauchy-Schwartz inequality; and the last inequality comes from the fact that $\|V_{P,r}^\pi\|_\infty \leqslant \frac{1}{1-\gamma}$. Thus, we have

$$\left\|\int_{\mathcal{A}}g(\cdot,a)\,\mathrm{d}a\right\|_2^2 = \int_{\mathcal{S}}\left(\int_{\mathcal{A}}g(s,a)\,\mathrm{d}a\right)^2\mathrm{d}s$$

$$\leqslant \frac{d}{(1-\gamma)^2} \int_{\mathcal{S}} \left( \int_{\mathcal{A}} \|\phi^*(s,a)\| \, \mathrm{d}a + \int_{\mathcal{A}} \|\widehat{\phi}_n(s,a)\| \, \mathrm{d}a \right)^2 \mathrm{d}s$$

$$\leqslant \frac{4d^2}{(1-\gamma)^2},$$

where the last inequality is due to Assumption 2 and the fact that $(a+b)^2 \leqslant 2(a^2+b^2)$. Invoking Lemma 7, we have that

$$\mathbb{E}_{(s,a)\sim d_{\widehat{P}_n}^{\pi}} \{g(s,a)\} \leqslant \sqrt{(1-\gamma)|\mathcal{A}|\mathbb{E}_{s\sim\rho_n, a\sim\mathcal{U}(\mathcal{A})}\{g^2(s,a)\}}$$

$$+ \gamma\sqrt{n|\mathcal{A}|\mathbb{E}_{s\sim\rho_n', a\sim\mathcal{U}(\mathcal{A})}\{g^2(s,a)\} + \frac{4d^2}{(1-\gamma)^2}n\zeta_n + \frac{4\lambda_n d}{(1-\gamma)^2}} \cdot \mathbb{E}_{(\tilde{s},\tilde{a})\sim d_{\widehat{P}_n}^{\pi}} \left[ \left\| \widehat{\phi}_n(\tilde{s},\tilde{a}) \right\|_{\Sigma_{\rho_n \times \mathcal{U}, \widehat{\phi}_n}^{-1}} \right].$$

Note that

$$\mathbb{E}_{s\sim\rho_n, a\sim\mathcal{U}(\mathcal{A})}\{g^2(s,a)\}$$

$$= \mathbb{E}_{s\sim\rho_n, a\sim\mathcal{U}(\mathcal{A})} \left( \int_{\mathcal{S}} \left( P(s'|s,a) - \widehat{P}_n(s'|s,a) \right) V_{P,r}^{\pi}(s') \, \mathrm{d}s' \right)^2$$

$$\leqslant \mathbb{E}_{s\sim\rho_n, a\sim\mathcal{U}(\mathcal{A})} \left\| P(\cdot|s,a) - \widehat{P}_n(\cdot|s,a) \right\|_2^2 \left\| V_{P,r}^{\pi} \right\|_2^2$$

$$\leqslant 2d \left( 1 + \frac{d\gamma^2}{(1-\gamma)^2} \right) \zeta_n,$$

where the first inequality is due to the Hölder's inequality and the last inequality is due to Lemma 6. With the selected hyperparameters and Lemma 8, we conclude the proof. $\square$

To further provide the regret bound, we need the following analog of Lemma 7. Note that, here we don't require $\rho_n'$.

**Lemma 10** (One-step back inequality for the true model). *Assume $g : \mathcal{S} \times \mathcal{A} \to \mathbb{R}$ satisfies that $\|g\|_\infty \leqslant B_\infty$, then we have that*

$$\left| \mathbb{E}_{(s,a)\sim d_P^{\pi_n}} \{g(s,a)\} \right| \leqslant \sqrt{(1-\gamma)|\mathcal{A}|\mathbb{E}_{s\sim\rho_n, a\sim\mathcal{U}(\mathcal{A})}\{g^2(s,a)\}}$$

$$+ \sqrt{n\gamma|\mathcal{A}|\mathbb{E}_{s\sim\rho_n, a\sim\mathcal{U}(\mathcal{A})}\{g^2(s,a)\} + \lambda_n\gamma^2 B_\infty^2 d} \cdot \mathbb{E}_{(\tilde{s},\tilde{a})\sim d_P^{\pi_n}} \left[ \|\phi^*(\tilde{s},\tilde{a})\|_{\Sigma_{\rho_n,\phi^*}^{-1}} \right].$$

*Proof.* Note that

$$\mathbb{E}_{(s,a)\sim d_P^{\pi_n}} \{g(s,a)\}$$

$$= \gamma\mathbb{E}_{(\tilde{s},\tilde{a})\sim d_P^{\pi_n}, s\sim P(\cdot|\tilde{s},\tilde{a}), a\sim\pi(\cdot|s)}\{g(s,a)\} + (1-\gamma)\mathbb{E}_{s\sim\rho, a\sim\pi(\cdot|s)}\{g(s,a)\}.$$

For the second term, note that $d_P^{\pi}(s) \geqslant (1-\gamma)\rho(s)$, hence

$$(1-\gamma)\mathbb{E}_{s\sim\rho, a\sim\pi(\cdot|s)}\{g(s,a)\}$$

$$\leqslant (1-\gamma)\sqrt{\mathbb{E}_{s\sim\rho, a\sim\pi(\cdot|s)}\{g^2(s,a)\}}$$

$$= (1-\gamma)\sqrt{\mathbb{E}_{s\sim\rho_n, a\sim\mathcal{U}(\mathcal{A})}\left\{ \frac{\rho(s)\pi(a|s)|\mathcal{A}|}{\rho_n(s)}g^2(s,a) \right\}}$$

$$\leqslant \sqrt{(1-\gamma)|\mathcal{A}|\mathbb{E}_{s\sim\rho_n, a\sim\mathcal{U}(\mathcal{A})}\{g^2(s,a)\}}.$$

For the first term, we have that

$$\mathbb{E}_{(\tilde{s},\tilde{a})\sim d_P^{\pi_n}, s\sim P(\cdot|\tilde{s},\tilde{a}), a\sim\pi(\cdot|s)}\{g(s,a)\}$$

$$= \mathbb{E}_{(\tilde{s},\tilde{a})\sim d_P^{\pi_n}} \phi^*(\tilde{s},\tilde{a})^{\top} \left[ \iint_{\mathcal{S}\times\mathcal{A}} \mu^*(s)\pi(a|s)g(s,a) \, \mathrm{d}s \, \mathrm{d}a \right]$$

$$\leqslant \mathbb{E}_{(\tilde{s},\tilde{a})\sim d_P^{\pi_n}} \|\phi^*(\tilde{s},\tilde{a})\|_{\Sigma_{\rho_n,\phi^*}^{-1}} \left\|\int_{\mathcal{S}\times\mathcal{A}} \mu^*(s)\pi(a|s)g(s,a)\,\mathrm{d}s\,\mathrm{d}a\right\|_{\Sigma_{\rho_n,\phi^*}},$$

where for the inequality we use the generalized Cauchy-Schwartz inequality. Note

$$\left\|\int_{\mathcal{S}\times\mathcal{A}} \mu^*(s)\pi(a|s)g(s,a)\,\mathrm{d}s\,\mathrm{d}a\right\|_{\Sigma_{\rho_n,\phi^*}}^2$$

$$=n\mathbb{E}_{(\tilde{s},\tilde{a})\sim\rho_n}\left[\left(\int_{\mathcal{S}\times\mathcal{A}} P(s|\tilde{s},\tilde{a})\pi(a|s)g(s,a)\,\mathrm{d}s\,\mathrm{d}a\right)^2\right] + \lambda_n\left\|\int_{\mathcal{S}\times\mathcal{A}} \mu^*(s)\pi(a|s)g(s,a)\,\mathrm{d}s\,\mathrm{d}a\right\|^2$$

$$\leqslant n\mathbb{E}_{(\tilde{s},\tilde{a})\sim\rho_n, s\sim P(\cdot|s,a), a\sim\pi(\cdot|s)}\{g^2(s,a)\} + \lambda_n B_\infty^2 d,$$

where in the last inequality we use Jensen's inequality. Note that

$$\mathbb{E}_{(\tilde{s},\tilde{a})\sim\rho_n, s\sim P(\cdot|s,a), a\sim\pi(\cdot|s)}\{g^2(s,a)\}$$

$$\leqslant \frac{1}{\gamma}\mathbb{E}_{(\tilde{s},\tilde{a})\sim\rho_n, s\sim P^*(\cdot|s,a), a\sim\pi(\cdot|s)}\{g^2(s,a)\}$$

$$\leqslant \frac{|\mathcal{A}|}{\gamma}\mathbb{E}_{s\sim\rho_n, a\sim\mathcal{U}(\mathcal{A})}\{g^2(s,a)\}$$

Substituting this back, we obtain the desired result. $\qquad\square$

We also need the following properties on the bonus and the value function when we plan on the learned model with the bonus.

**Lemma 11** (Norm of the Bonus). *We have that*

$$\|b_n(s,a)\|_\infty \leqslant \frac{\alpha_n}{\sqrt{\lambda_n}} \lesssim \frac{\sqrt{d|\mathcal{A}|}}{1-\gamma}, \quad \left\|\int_{\mathcal{A}} b_n(\cdot,a)\,\mathrm{d}a\right\| \leqslant \frac{\alpha_n\sqrt{d}}{\sqrt{\lambda_n}} \lesssim \frac{d\sqrt{|\mathcal{A}|}}{1-\gamma}.$$

*Proof.* Note that, $\widehat{\Sigma}_{n,\widehat{\phi}_n} \gtrsim \lambda_n I$, and as a result, we have $\left\|\widehat{\Sigma}_{n,\widehat{\phi}_n}^{-1}\right\|_{\mathrm{op}} \leqslant \frac{1}{\lambda_n}$. Recall $b_n(s,a) = \alpha_n\left\|\widehat{\phi}_n(s,a)\right\|_{\widehat{\Sigma}_{n,\widehat{\phi}_n}^{-1}}$, we know

$$b_n^2(s,a) = \alpha_n^2\widehat{\phi}_n(s,a)\widehat{\Sigma}_{n,\widehat{\phi}_n}^{-1}\widehat{\phi}_n(s,a) \leqslant \frac{\alpha_n^2\|\widehat{\phi}_n(s,a)\|_2^2}{\lambda_n} \leqslant \frac{\alpha_n^2}{\lambda_n},$$

as well as

$$\left\|\int_{\mathcal{A}} b_n(\cdot,a)\,\mathrm{d}a\right\|^2$$

$$=\alpha_n^2\left\|\int_{\mathcal{A}} \|\widehat{\phi}_n(\cdot,a)\|_{\widehat{\Sigma}_n^{-1},\widehat{\phi}_n}\,\mathrm{d}a\right\|^2$$

$$=\alpha_n^2\int_{\mathcal{S}}\left(\int_{\mathcal{A}} \|\widehat{\phi}_n(\cdot,a)\|_{\widehat{\Sigma}_n^{-1},\widehat{\phi}_n}\,\mathrm{d}a\right)^2\mathrm{d}s$$

$$\leqslant \frac{\alpha_n^2}{\lambda_n}\int_{\mathcal{S}}\left(\int_{\mathcal{A}} \|\widehat{\phi}_n(s,a)\|\,\mathrm{d}a\right)^2\mathrm{d}s$$

$$\leqslant \frac{\alpha_n^2 d}{\lambda_n},$$

Combined with the fact that $\frac{\alpha_n}{\sqrt{\lambda_n}} = \Theta\left(\frac{\sqrt{d|\mathcal{A}|}}{1-\gamma}\right)$, we conclude the proof. $\qquad\square$

**Lemma 12** ($L_2$ norm of $V_{\widehat{P}_n,r+b_n}^\pi$). *For any policy $\pi$, we have that*

$$\left\|V_{\widehat{P}_n,r+b_n}^\pi\right\| \leqslant \sqrt{3d + \frac{3\alpha_n^2 d}{\lambda_n} + \frac{3d^2\gamma^2\left(1+\frac{\alpha_n}{\sqrt{\lambda_n}}\right)^2}{(1-\gamma)^2}} \lesssim \frac{d^{1.5}\sqrt{|\mathcal{A}|}}{(1-\gamma)^2}$$

*Proof.* We have

$$\left\| V^{\pi}_{\widehat{P}_n, r+b_n} \right\|^2$$

$$= \int_{\mathcal{S}} \left( V^{\pi}_{\widehat{P}_n, r+b_n}(s) \right)^2 \mathrm{d}s$$

$$= \int_{\mathcal{S}} \left( \int_{\mathcal{A}} \pi(a|s) \left( r(s,a) + b_n(s,a) + \gamma Q^{\pi}_{\widehat{P}_n, r+b_n}(s,a) \right) \mathrm{d}a \right)^2 \mathrm{d}s$$

$$\leqslant \int_{\mathcal{S}} \left( \int_{\mathcal{A}} \left( r(s,a) + b_n(s,a) + \gamma Q^{\pi}_{\widehat{P}_n, r+b_n}(s,a) \right) \mathrm{d}a \right)^2 \mathrm{d}s$$

$$\leqslant 3 \int_{\mathcal{S}} \left( \int_{\mathcal{A}} [r(s,a)]^2 \, \mathrm{d}a \right) \mathrm{d}s + 3 \int_{\mathcal{S}} \left( \int_{\mathcal{A}} \left[ \alpha_n \left\| \widehat{\phi}_n(s,a) \right\|_{\Sigma^{-1}_{\rho_n, \widehat{\phi}_n}} \mathrm{d}a \right]^2 \mathrm{d}a \right) \mathrm{d}s$$

$$\quad + 3\gamma^2 \int_{\mathcal{S}} \left( \int_{\mathcal{A}} \left[ Q^{\pi}_{\widehat{P}_n, r+b_n}(s,a) \right]^2 \mathrm{d}a \right)^2 \mathrm{d}s$$

$$\leqslant 3d + \frac{3\alpha_n^2 d}{\lambda_n} + \frac{3d^2\gamma^2 \left( 1 + \frac{\alpha_n}{\sqrt{\lambda_n}} \right)^2}{(1-\gamma)^2}$$

$$\lesssim \frac{d^3 |\mathcal{A}|}{(1-\gamma)^4},$$

which concludes the proof. $\qquad\square$

Now we are ready to prove the following regret bounds and obtain the final PAC guarantee.

**Lemma 13** (Regret). *With probability at least $1 - \delta$, we have that*

$$\sum_{n=1}^{N} V^{\pi^*}_{P,r} - V^{\pi_n}_{P,r} \lesssim \sqrt{\frac{Nd^4|\mathcal{A}|^2 \log(N|\mathcal{F}|/\delta)}{(1-\gamma)^6}} \log \left( 1 + \frac{N}{d^2 \log(N|\mathcal{F}|/\delta)} \right).$$

*Proof.* Standard decomposition shows

$$V^{\pi^*}_{P,r} - V^{\pi_n}_{P,r}$$

$$\leqslant V^{\pi^*}_{\widehat{P}_n, r+b_n} + \sqrt{\frac{2|\mathcal{A}|d \left( 1 + \frac{\gamma^2 d}{(1-\gamma)^2} \right) \zeta_n}{(1-\gamma)}} - V^{\pi_n}_{P,r}$$

$$\leqslant V^{\pi_n}_{\widehat{P}_n, r+b_n} - V^{\pi_n}_{P,r} + \sqrt{\frac{2|\mathcal{A}|d \left( 1 + \frac{\gamma^2 d}{(1-\gamma)^2} \right) \zeta_n}{(1-\gamma)}}$$

$$\leqslant \frac{1}{1-\gamma} \mathbb{E}_{(s,a) \sim d^{\pi_n}_P} \left[ b_n(s,a) + \gamma \mathbb{E}_{\widehat{P}_n(s'|s,a)}[V^{\pi_n}_{\widehat{P}_n, r+b_n}(s')] - \gamma \mathbb{E}_{P(s'|s,a)}[V^{\pi_n}_{\widehat{P}_n, r+b_n}(s')] \right]$$

$$\quad + \sqrt{\frac{2|\mathcal{A}|d \left( 1 + \frac{\gamma^2 d}{(1-\gamma)^2} \right) \zeta_n}{(1-\gamma)}}.$$

Applying Lemma 10 to $\mathbb{E}_{(s,a) \sim d^{\pi_n}_P} \{b_n(s,a)\}$, we have that

$$\mathbb{E}_{(s,a) \sim d^{\pi_n}_P} \{b_n(s,a)\}$$

$$\leqslant \sqrt{(1-\gamma)|\mathcal{A}| \mathbb{E}_{s \sim \rho_n, a \sim \mathcal{U}(\mathcal{A})} \{b_n^2(s,a)\}}$$

$$\quad + \sqrt{n\gamma |\mathcal{A}| \mathbb{E}_{s \sim \rho_n, a \sim \mathcal{U}(\mathcal{A})} \{b_n^2(s,a)\} + \gamma^2 \alpha_n^2 d \mathbb{E}_{(\tilde{s}, \tilde{a}) \sim d^{\pi_n}_P} \left[ \|\phi^*(\tilde{s}, \tilde{a})\|_{\Sigma^{-1}_{\rho_n, \phi^*}} \right]}.$$

Note that

$$\mathbb{E}_{s\sim\rho_n,a\sim\mathcal{U}(\mathcal{A})}\left\|\widehat{\phi}_n(s,a)\right\|^2_{\Sigma^{-1}_{\rho_n,\widehat{\phi}_n}}$$

$$=\mathbb{E}_{s\sim\rho_n,a\sim\mathcal{U}(\mathcal{A})}\left[\widehat{\phi}_n(s,a)^\top\Sigma^{-1}_{\rho_n,\widehat{\phi}_n}\widehat{\phi}_n(s,a)\right]$$

$$=\mathrm{Tr}\left(\mathbb{E}_{s\sim\rho_n,a\sim\mathcal{U}(\mathcal{A})}\left[\widehat{\phi}_n(s,a)\widehat{\phi}_n(s,a)^\top\right]\left(n\mathbb{E}_{s\sim\rho_n,a\sim\mathcal{U}(\mathcal{A})}\left[\widehat{\phi}_n(s,a)\widehat{\phi}_n(s,a)^\top\right]+\lambda_nI\right)^{-1}\right)$$

$$\leqslant\frac{d}{n},$$

hence, with the concentration of the bonus, we have

$$\mathbb{E}_{(s,a)\sim d_P^{\pi_n}}\{b_n(s,a)\}\lesssim\sqrt{\frac{(1-\gamma)\alpha_n^2d|\mathcal{A}|}{n}}+\sqrt{\gamma\alpha_n^2d|\mathcal{A}|+\gamma^2\alpha_n^2d}\cdot\mathbb{E}_{(\tilde{s},\tilde{a})\sim d_P^{\pi_n}}\left[\|\phi^*(\tilde{s},\tilde{a})\|_{\Sigma^{-1}_{\rho_n,\phi^*}}\right].$$

We then consider the remaining term. With a slightly abuse of notation, define $g(s,a):=\left|\mathbb{E}_{\widehat{P}_n(s'|s,a)}V^{\pi_n}_{\widehat{P}_n,r+b_n}(s')-\mathbb{E}_{P(s'|s,a)}V^{\pi_n}_{\widehat{P}_n,r+b_n}(s')\right|$. With Hölder's inequality, we know that $\|g(s,a)\|_\infty\leqslant 2\left\|V^{\pi_n}_{\widehat{P}_n,r+b_n}\right\|_\infty\leqslant\frac{2\left(1+\frac{\alpha_n}{\sqrt{\lambda_n}}\right)}{1-\gamma}\lesssim\frac{\sqrt{d|\mathcal{A}|}}{(1-\gamma)^2}$. Applying Lemma 10 to $\mathbb{E}_{(s,a)\sim d_P^{\pi_n}}\{g(s,a)\}$, we have that

$$\mathbb{E}_{(s,a)\sim d_P^{\pi_n}}\{g(s,a)\}$$

$$\leqslant\sqrt{(1-\gamma)|\mathcal{A}|\mathbb{E}_{s\sim\rho_n,a\sim\mathcal{U}(\mathcal{A})}\{g^2(s,a)\}}$$

$$+\sqrt{n\gamma|\mathcal{A}|\mathbb{E}_{s\sim\rho_n,a\sim\mathcal{U}(\mathcal{A})}\{g^2(s,a)\}+\frac{4\gamma^2d\left(\sqrt{\lambda_n}+\alpha_n\right)^2}{(1-\gamma)^2}\mathbb{E}_{(\tilde{s},\tilde{a})\sim d_P^{\pi_n}}\left[\|\phi^*(\tilde{s},\tilde{a})\|_{\Sigma^{-1}_{\rho_n,\phi^*}}\right]}.$$

Note that

$$\mathbb{E}_{s\sim\rho_n,a\sim\mathcal{U}(\mathcal{A})}\{g^2(s,a)\}$$

$$=\mathbb{E}_{s\sim\rho_n,a\sim\mathcal{U}(\mathcal{A})}\left[\left(\int_\mathcal{S}\left(\widehat{P}_n(s'|s,a)-P(s'|s,a)\right)V^{\pi_n}_{\widehat{P}_n,r+b_n}(s')\right)^2\right]$$

$$\leqslant\mathbb{E}_{s\sim\rho_n,a\sim\mathcal{U}(\mathcal{A})}\left[\left\|\widehat{P}_n(\cdot|s,a)-P(\cdot|s,a)\right\|^2\left\|V^{\pi_n}_{\widehat{P}_n,r+b_n}\right\|^2\right]$$

$$\leqslant 3d\left(1+\frac{\alpha_n^2}{\lambda_n}+\frac{d\gamma^2\left(1+\frac{\alpha_n}{\sqrt{\lambda_n}}\right)^2}{(1-\gamma)^2}\right)\zeta_n\lesssim\frac{d^3|\mathcal{A}|\zeta_n}{(1-\gamma)^4}$$

Hence,

$$\mathbb{E}_{(s,a)\sim d_P^{\pi_n}}\{g(s,a)\}\lesssim\sqrt{(1-\gamma)d^3|\mathcal{A}|^2\zeta_n}$$

$$+\sqrt{\frac{d^3|\mathcal{A}|^2n\zeta_n}{(1-\gamma)^4}+\frac{d^3|\mathcal{A}|n\zeta_n}{(1-\gamma)^4}\mathbb{E}_{(\tilde{s},\tilde{a})\sim d_P^{\pi_n}}\left[\|\phi^*(\tilde{s},\tilde{a})\|_{\Sigma^{-1}_{\rho_n,\phi^*}}\right]}.$$

Finally, with Lemma 20 and notice that $\lambda_1\leqslant\lambda_2\leqslant\cdots\leqslant\lambda_N$, we have that

$$\sum_{n=1}^N\mathbb{E}_{(\tilde{s},\tilde{a})\sim d_P^{\pi_n}}\|\phi^*(\tilde{s},\tilde{a})\|_{\Sigma^{-1}_{\rho_n,\phi^*}}$$

$$\leqslant\sqrt{N\mathrm{Tr}\left(\left(\mathbb{E}_{(\tilde{s},\tilde{a})\sim d_P^{\pi_n}}\phi^*(\tilde{s},\tilde{a})\left(\phi^*(\tilde{s},\tilde{a})\right)^\top\right)\Sigma^{-1}_{\rho_n,\phi^*}\right)}$$

$$\leqslant\sqrt{Nd\log\frac{\lambda_N+N}{\lambda_1}}$$

Combine the previous terms and take the dominating terms out, we have that

$$\sum_{n=1}^{N} V_{P,r}^{\pi^*} - V_{P,r}^{\pi_n} \lesssim \sqrt{\frac{Nd^4|\mathcal{A}|^2 \log(N|\mathcal{F}|/\delta)}{(1-\gamma)^6} \log\left(1 + \frac{N}{d^2 \log(N|\mathcal{F}|/\delta)}\right)},$$

which concludes the proof. $\qquad\square$

**Theorem 14** (PAC Guarantee). *After interacting with the environments for $N = \widetilde{\Theta}\left(\frac{d^4|\mathcal{A}|^2}{(1-\gamma)^6\epsilon^2}\right)$ episodes, we can obtain an $\epsilon$-optimal policy with high probability. Furthermore, with high probability, for each episode, we can terminate within $\widetilde{\Theta}(1/(1-\gamma))$ steps.*

*Proof.* It directly follows from the standard regret to PAC reduction. See Jin et al. (2018); Uehara et al. (2022) for the detail. $\qquad\square$

### D.3 PAC BOUNDS FOR OFFLINE REINFORCEMENT LEARNING

**Proof Sketch** Similar to the online counterpart, our proof for offline setting is organized as follows:

- We show the policy obtained by planning on the learned model with additional penalty lower bound the optimal value up to some error term (Lemma 16), with the help of an analog of the one-step back inequality for the learned model in the offline setting (Lemma 15) based on Theorem 5.
- We then show the PAC guarantee (Theorem 18) with an analog of the one-step back inequality for the true model in the offline setting (Lemma 17).

We first prove the analog of Lemma 7 in the offline setting.

**Lemma 15** (One-step back inequality for the learned model in the offline setting). *Let $\omega = \max_{s,a}\{1/\pi_b(a|s)\}$. Assume $g : \mathcal{S} \times \mathcal{A} \to \mathbb{R}$ satisfies that $\|g\|_\infty \leqslant B_\infty$, $\|\int_{\mathcal{A}} g(\cdot, a)\,\mathrm{d}a\|_2 \leqslant B_2$, then we have that*

$$\left|\mathbb{E}_{(s,a)\sim d_{\widehat{P}}^{\pi}}\{g(s,a)\}\right| \leqslant \sqrt{(1-\gamma)\omega\mathbb{E}_{(s,a)\sim\rho_b}\{g^2(s,a)\}}$$

$$+ \gamma\sqrt{n\omega\mathbb{E}_{(s,a)\sim\rho_b}\{g^2(s,a)\} + B_2^2 n\zeta_n + \lambda_n B_\infty^2 d} \cdot \mathbb{E}_{\tilde{s},\tilde{a}\sim d_{\widehat{P}}^{\pi}}\left[\left\|\widehat{\phi}(\tilde{s},\tilde{a})\right\|_{\Sigma_{\rho_b,\phi}^{-1}}\right].$$

*Proof.* Note that

$$\mathbb{E}_{(s,a)\sim d_{\widehat{P}}^{\pi}}\{g(s,a)\} = \gamma\mathbb{E}_{(\tilde{s},\tilde{a})\sim d_{\widehat{P}}^{\pi}, s\sim\widehat{P}(\cdot|\tilde{s},\tilde{a}), a\sim\pi(\cdot|s)}\{g(s,a)\} + (1-\gamma)\mathbb{E}_{s\sim\rho, a\sim\pi(\cdot|s)}\{g(s,a)\}.$$

For the second term, we have that

$$(1-\gamma)\mathbb{E}_{s\sim\rho, a\sim\pi(\cdot|s)}\{g(s,a)\}$$
$$\leqslant (1-\gamma)\sqrt{\mathbb{E}_{s\sim\rho, a\sim\pi(\cdot|s)}\{g^2(s,a)\}}$$
$$= (1-\gamma)\sqrt{\mathbb{E}_{s\sim\rho_b, a\sim\pi_b(\cdot|s)}\left\{\frac{\rho(s)\pi(a|s)}{\rho_b(s)\pi(a|s)}g^2(s,a)\right\}}$$
$$\leqslant \sqrt{\omega(1-\gamma)|\mathcal{A}|\mathbb{E}_{s\sim\rho_b, a\sim\pi_b(\cdot|s)}g^2(s,a)}.$$

For the first term, we have that

$$\mathbb{E}_{(\tilde{s},\tilde{a})\sim d_{\widehat{P}}^{\pi}, s\sim\widehat{P}(\cdot|\tilde{s},\tilde{a}), a\sim\pi(\cdot|s)}\{g(s,a)\}$$
$$= \mathbb{E}_{(\tilde{s},\tilde{a})\sim d_{\widehat{P}}^{\pi}}\widehat{\phi}(\tilde{s},\tilde{a})^\top\left[\int_{\mathcal{S}\times\mathcal{A}}\widehat{\mu}(s)\pi(a|s)g(s,a)\right]$$
$$\leqslant \mathbb{E}_{(\tilde{s},\tilde{a})\sim d_{\widehat{P}}^{\pi}}\left\|\widehat{\phi}(\tilde{s},\tilde{a})\right\|_{\Sigma_{\rho_b,\widehat{\phi}}^{-1}}\left\|\int_{\mathcal{S}\times\mathcal{A}}\widehat{\mu}(s)\pi(a|s)g(s,a)\,\mathrm{d}s\,\mathrm{d}a\right\|_{\Sigma_{\rho_b,\widehat{\phi}}}.$$

Note that

$$
\left\| \int_{\mathcal{S} \times \mathcal{A}} \widehat{\mu}(s) \pi(a|s) g(s,a) \, \mathrm{d}s \, \mathrm{d}a \right\|^2_{\Sigma_{\rho_b, \widehat{\phi}}}
$$

$$
= n \mathbb{E}_{(\tilde{s}, \tilde{a}) \sim \rho_b} \left[ \left( \int_{\mathcal{S} \times \mathcal{A}} \widehat{P}_n(s|\tilde{s}, \tilde{a}) \pi(a|s) g(s,a) \, \mathrm{d}s \, \mathrm{d}a \right)^2 \right] + \lambda \left\| \int_{\mathcal{S} \times \mathcal{A}} \widehat{\mu}(s) \pi(a|s) g(s,a) \right\|^2
$$

$$
\leqslant 2n \mathbb{E}_{(\tilde{s}, \tilde{a}) \sim \rho_b} \left[ \left( \int_{\mathcal{S} \times \mathcal{A}} P(s|\tilde{s}, \tilde{a}) \pi(a|s) g(s,a) \, \mathrm{d}s \, \mathrm{d}a \right)^2 \right]
$$

$$
+ 2n \mathbb{E}_{(\tilde{s}, \tilde{a}) \sim \rho_b} \left[ \left( \int_{\mathcal{S} \times \mathcal{A}} \left( \widehat{P}(s|\tilde{s}, \tilde{a}) - P(s|\tilde{s}, \tilde{a}) \right) \pi(a|s) g(s,a) \, \mathrm{d}s \, \mathrm{d}a \right)^2 \right] + \lambda B_\infty^2 d.
$$

With Jensen's inequality, we have

$$
\mathbb{E}_{(\tilde{s}, \tilde{a}) \sim \rho_b} \left[ \left( \int_{\mathcal{S} \times \mathcal{A}} P(s|\tilde{s}, \tilde{a}) \pi(a|s) g(s,a) \, \mathrm{d}s \, \mathrm{d}a \right)^2 \right]
$$

$$
\leqslant \mathbb{E}_{(\tilde{s}, \tilde{a}) \sim \rho_b, s \sim P(\cdot|s,a), a \sim \pi(\cdot|s)} \{ g^2(s,a) \}
$$

$$
\leqslant \frac{\omega}{\gamma} \mathbb{E}_{(s,a) \sim \rho_b} \{ g^2(s,a) \}.
$$

On the other hand,

$$
\mathbb{E}_{(\tilde{s}, \tilde{a}) \sim \rho_b} \left[ \left( \int_{\mathcal{S} \times \mathcal{A}} \left( \widehat{P}(s|\tilde{s}, \tilde{a}) - P(s|\tilde{s}, \tilde{a}) \right) \pi(a|s) g(s,a) \, \mathrm{d}s \, \mathrm{d}a \right)^2 \right]
$$

$$
\leqslant \mathbb{E}_{(\tilde{s}, \tilde{a}) \sim \rho_b} \left[ \left\| \widehat{P}(\cdot|\tilde{s}, \tilde{a}) - P(\cdot|\tilde{s}, \tilde{a}) \right\|^2_2 \left\| \int_{\mathcal{A}} \pi(a|\cdot) g(\cdot, a) \, \mathrm{d}a \right\|^2_2 \right]
$$

$$
\leqslant \mathbb{E}_{(\tilde{s}, \tilde{a}) \sim \rho_b} \left[ \left\| \widehat{P}(\cdot|\tilde{s}, \tilde{a}) - P(\cdot|\tilde{s}, \tilde{a}) \right\|^2_2 \left\| \int_{\mathcal{A}} g(\cdot, a) \, \mathrm{d}a \right\|^2_2 \right]
$$

$$
\leqslant B_2^2 \zeta_n,
$$

where the last inequality is due to Theorem 5. Substituting this back, we obtain the desired result. $\quad\square$

**Lemma 16** (Pessimism). *Let* $\omega = \max_{s,a} \{ \pi_b(a|s) \}$,

$$
\alpha_n = \Theta \left( \frac{d \sqrt{\omega \zeta_n}}{1 - \gamma} \right),
$$

$$
\lambda = \Theta(d \log(|\mathcal{F}|/\delta)),
$$

*then we have*

$$
V^\pi_{\widehat{P}, r-b} \leqslant V^\pi_{P,r} + \sqrt{ \frac{2 \omega d \left( 1 + \frac{\gamma^2 d}{(1-\gamma)^2} \right) \zeta_n}{(1 - \gamma)} }.
$$

*Proof.* With the simulation Lemma (*i.e.*, Lemma 19), we have

$$
V^\pi_{\widehat{P}_n, r-b} - V^\pi_{P,r}
$$

$$
= \frac{1}{1 - \gamma} \mathbb{E}_{(s,a) \sim d^\pi_{\widehat{P}_n}} \left[ -b(s,a) + \gamma \left[ \mathbb{E}_{\widehat{P}_n(s'|s,a)} \left[ V^\pi_{P,r}(s') \right] - \mathbb{E}_{P(s'|s,a)} \left[ V^\pi_{P,r}(s') \right] \right] \right].
$$

Consider $g(s,a) := \left| \left[ \mathbb{E}_{\widehat{P}_n(s'|s,a)} \left[ V^\pi_{P,r}(s') \right] - \mathbb{E}_{P(s'|s,a)} \left[ V^\pi_{P,r}(s') \right] \right] \right|$. With Hölder's inequality, $\|g\|_\infty \leqslant \frac{2}{1-\gamma}$. Furthermore, with the derivation in the proof of Lemma 9, we know $\left\| \int_{\mathcal{A}} g(\cdot, a) \, \mathrm{d}a \right\|_2 \leqslant \frac{\sqrt{2}d}{1-\gamma}$. Applying Lemma 15, we have that

$$
\mathbb{E}_{(s,a) \sim d^\pi_{\widehat{P}}} \{ g(s,a) \} \leqslant \sqrt{ (1 - \gamma) \omega \mathbb{E}_{(s,a) \sim \rho_b} \{ g^2(s,a) \} }
$$

$$+ \gamma \sqrt{n\omega \mathbb{E}_{(s,a)\sim\rho_b}\{g^2(s,a)\} + \frac{2d^2}{(1-\gamma)^2}\log(|\mathcal{F}|/\delta) + \frac{4\lambda_n d}{(1-\gamma)^2}} \cdot \mathbb{E}_{(\tilde{s},\tilde{a})\sim d_{\widehat{P}}^\pi}\left[\left\|\widehat{\phi}(\tilde{s},\tilde{a})\right\|_{\Sigma_{\rho_b,\widehat{\phi}}^{-1}}\right].$$

With Lemma 6, we know

$$\mathbb{E}_{(s,a)\sim\rho_b}\{g^2(s,a)\} \leqslant 2d(1 + \frac{d\gamma^2}{(1-\gamma)^2})\zeta_n.$$

Then, with the selected hyperparameters and Lemma 8, we conclude the proof. $\square$

**Lemma 17** (One-step back inequality for the true model in the offline setting). *Let* $\omega = \max_{s,a}\{\pi_b(a|s)\}$, *assume* $g : \mathcal{S} \times \mathcal{A} \to \mathbb{R}$ *satisfies* $\|g\|_\infty \leqslant B_\infty$, *then we have*

$$\left|\mathbb{E}_{(s,a)\sim d_P^\pi}\{g(s,a)\}\right| \leqslant \sqrt{(1-\gamma)\omega\mathbb{E}_{(s,a)\sim\rho_b}\{g^2(s,a)\}}$$

$$+ \sqrt{n\gamma\omega\mathbb{E}_{(s,a)\sim\rho_b}\{g^2(s,a)\} + \lambda\gamma^2 B_\infty^2 d} \cdot \mathbb{E}_{(\tilde{s},\tilde{a})\sim d_P^\pi}\left[\|\phi^*(\tilde{s},\tilde{a})\|_{\Sigma_{\rho_b,\sigma^*}^{-1}}\right].$$

*Proof.* The proof is identical to the proof of Lemma 7. $\square$

We now provide the PAC guarantee for the offline setting.

**Theorem 18** (PAC Guarantee). *With probability* $1-\delta$, *$\forall$ baseline policy $\pi$ including history-dependent non-Markovian policies, we have that*

$$V_{P,r}^\pi - V_{P,r}^{\widehat{\pi}} \lesssim \sqrt{\frac{\omega^2 d^4 C_\pi^* \log(|\mathcal{F}|/\delta)}{(1-\gamma)^6}},$$

*where $C_\pi^*$ is the relative conditional number under $\phi^*$, defined as*

$$C_\pi^* := \sup_{x\in\mathbb{R}^d} \frac{x^\top \mathbb{E}_{(s,a)\sim d_P^\pi}[\phi^*(s,a)\phi^*(s,a)^\top]x}{x^\top \mathbb{E}_{(s,a)\sim\rho_b}[\phi^*(s,a)\phi^*(s,a)^\top]x}.$$

*Proof.* Standard decomposition shows

$$V_{P,r}^\pi - V_{P,r}^{\widehat{\pi}}$$

$$\leqslant V_{P,r}^\pi - V_{\widehat{P},r-b}^{\widehat{\pi}} + \sqrt{\frac{2\omega d\left(1 + \frac{\gamma^2 d}{(1-\gamma^2)}\right)\zeta_n)}{(1-\gamma)}}$$

$$\leqslant V_{P,r}^\pi - V_{\widehat{P},r-b}^\pi + \sqrt{\frac{2\omega d\left(1 + \frac{\gamma^2 d}{(1-\gamma^2)}\right)\zeta_n}{(1-\gamma)}}$$

$$= \mathbb{E}_{(s,a)\sim d_P^\pi}\left[b(s,a) + \gamma\mathbb{E}_{P(s'|s,a)}\left[V_{\widehat{P},r-b}^\pi(s')\right] - \gamma\mathbb{E}_{\widehat{P}(s'|s,a)}\left[V_{\widehat{P},r-b}^\pi(s')\right]\right]$$

$$+ \sqrt{\frac{2\omega d\left(1 + \frac{\gamma^2 d}{(1-\gamma^2)}\right)\zeta_n}{(1-\gamma)}}.$$

With Lemma 17 and the identical method used in the proof of Lemma 13, we have that

$$\mathbb{E}_{(s,a)\sim d_P^\pi}\{b_n(s,a)\} \leqslant \sqrt{\frac{(1-\gamma)\alpha_n^2 d\omega}{n}} + \sqrt{\gamma\alpha_n^2 d\omega + \gamma^2\alpha_n^2 d} \cdot \mathbb{E}_{(\tilde{s},\tilde{a})\sim d_P^\pi}\left[\|\phi^*(\tilde{s},\tilde{a})\|_{\Sigma_{\rho_b,\phi^*}}\right].$$

Furthermore, define $g(s,a) := \left|\mathbb{E}_{\widehat{P}_n(s'|s,a)}V_{\widehat{P}_n,r+b_n}^{\pi_n}(s') - \mathbb{E}_{P(s'|s,a)}V_{\widehat{P}_n,r+b_n}^{\pi_n}(s')\right|$. With Lemma 17 and the identical method used in the proof of Lemma 13, we can obtain

$$\mathbb{E}_{(s,a)\sim d_P^\pi}\{g(s,a)\} \lesssim \sqrt{(1-\gamma)d^3\omega^2\zeta_n} + \sqrt{\frac{d^3\omega^2 n\zeta_n}{(1-\gamma)^4}}\mathbb{E}_{(\tilde{s},\tilde{a})\sim d_P^\pi}\left[\|\phi^*(\tilde{s},\tilde{a})\|_{\Sigma_{\rho_b,\phi^*}^{-1}}\right].$$

Finally, by the definition of $C^*$, we have that

$$\mathbb{E}_{(\tilde{s},\tilde{a})\sim d_P^\pi}\left[\|\phi^*(\tilde{s},\tilde{a})\|_{\Sigma_{\rho_b,\phi^*}^{-1}}\right]\leqslant\sqrt{\mathbb{E}_{(\tilde{s},\tilde{a})\sim d_P^\pi}\left[\|\phi^*(\tilde{s},\tilde{a})\|_{\Sigma_{\rho_b,\phi^*}^{-1}}^2\right]}$$

$$\leqslant\sqrt{C^*\mathbb{E}_{(\tilde{s},\tilde{a})\sim\rho_b}\left[\|\phi^*(\tilde{s},\tilde{a})\|_{\Sigma_{\rho_b,\phi^*}^{-1}}\right]}\leqslant\sqrt{\frac{C^*d}{n}}.$$

Combining the previous terms and taking the dominating terms out, we conclude the proof. $\quad\square$

# E   TECHNICAL LEMMAS

**Lemma 19** (Simulation Lemma). *With a slightly abuse of notation, we have*

$$V_{\widehat{P}_n,r+b}^\pi - V_{P,r}^\pi = \frac{1}{1-\gamma}\mathbb{E}_{(s,a)\sim d_P^\pi}\left[b(s,a)+\gamma\left[\mathbb{E}_{\widehat{P}_n(s'|s,a)}[V_{\widehat{P}_n,r+b}^\pi(s')]-\mathbb{E}_{P(s'|s,a)}[V_{\widehat{P}_n,r+b}^\pi(s')]\right]\right],$$

$$V_{\widehat{P}_n,r+b}^\pi - V_{P,r}^\pi = \frac{1}{1-\gamma}\mathbb{E}_{(s,a)\sim d_{\widehat{P}_n}^\pi}\left[b(s,a)+\gamma\left[\mathbb{E}_{\widehat{P}_n(s'|s,a)}[V_{P,r}^\pi(s')]-\mathbb{E}_{P(s'|s,a)}[V_{P,r}^\pi(s')]\right]\right].$$

*Proof.* Note that

$$\mathbb{E}_{s\sim d_P^\pi,a\sim\pi(\cdot|s)}[f(s,a)]$$
$$=(1-\gamma)\mathbb{E}_{s\sim\rho,a\sim\pi(\cdot|s)}[f(s,a)]+\gamma\mathbb{E}_{\tilde{s}\sim d_P^\pi,\tilde{a}\sim\pi(\cdot|\tilde{s}),s\sim P(\cdot|\tilde{s},\tilde{a}),\tilde{a}\sim\pi(\cdot|\tilde{s})}[f(s,a)].$$

Take $f=Q_{\widehat{P}_n,r+b}^\pi$, we have that

$$V_{\widehat{P}_n,r+b}^\pi = \mathbb{E}_{s\sim\rho,a\sim\pi(\cdot|s)}\left[Q_{\widehat{P}_n,r+b}^\pi(s,a)\right]$$
$$=\frac{1}{1-\gamma}\left(\mathbb{E}_{s\sim d_P^\pi,a\sim\pi(\cdot|s)}\left[Q_{\widehat{P}_n,r+b}^\pi(s,a)\right]-\gamma\mathbb{E}_{\tilde{s}\sim d_P^\pi,\tilde{a}\sim\pi(\cdot|\tilde{s}),s\sim P(\cdot|\tilde{s},\tilde{a}),\tilde{a}\sim\pi(\cdot|\tilde{s})}\left[Q_{\widehat{P}_n,r+b}^\pi(s,a)\right]\right)$$
$$=\frac{1}{1-\gamma}\mathbb{E}_{s\sim d_P^\pi,a\sim\pi(\cdot|s)}\left[Q_{\widehat{P}_n,r+b}^\pi(s,a)-\gamma\mathbb{E}_{s'\sim P(\cdot|s,a),a'\sim\pi(\cdot|s')}\left[Q_{\widehat{P}_n,r+b}^\pi(s',a')\right]\right].$$

Substitute back, we have that

$$V_{\widehat{P}_n,r+b}^\pi - V_{P,r}^\pi$$
$$=\frac{1}{1-\gamma}\mathbb{E}_{s\sim d_P^\pi,a\sim\pi(\cdot|s)}\left[Q_{\widehat{P}_n,r+b}^\pi(s,a)-\gamma\mathbb{E}_{s'\sim P(\cdot|s,a),a'\sim\pi(\cdot|s')}\left[Q_{\widehat{P}_n,r+b}^\pi(s',a')\right]-r(s,a)\right]$$
$$=\frac{1}{1-\gamma}\mathbb{E}_{s\sim d_P^\pi,a\sim\pi(\cdot|s)}\left[b(s,a)+\gamma\left[\mathbb{E}_{s'\sim\widehat{P}_n(\cdot|s,a)}[V_{\widehat{P},r+b}^\pi(s')]-\mathbb{E}_{s'\sim P(\cdot|s,a)}[V_{\widehat{P}_n,r+b}^\pi(s')]\right]\right].$$

The second equation can be obtained with a similar method, which concludes the proof. $\quad\square$

**Lemma 20** (Elliptical Potential Lemma). *Let $M_0=\lambda I_{d\times d}$, $M_n=M_{n-1}+G_n$ where $G_n$ is a symmetric positive definite matrix with $\|G_n\|_{op}\leqslant c$, then we have that*

$$\sum_{n=1}^N\text{Tr}(G_nM_n^{-1})\leqslant\log\det(M_N)-2d\log\lambda\leqslant d\log\left(1+\frac{Nc}{\lambda}\right).$$

*Proof.* By the concavity of $\log\det(\cdot)$ function and $\frac{d\log\det(X)}{dX}=(X^\top)^{-1}$, we know

$$\log\det(M_{n-1})\leqslant\log\det(M_n)+\text{Tr}(M_n^{-1}(M_{n-1}-M_n))$$
$$=\log\det(M_n)-\text{Tr}(M_n^{-1}G_n).$$

Telescoping, we can obtain the first inequality. For the second inequality, note that, with Jensen's inequality, we have

$$\log\det(M_n)=\sum_{i=1}^d\log\sigma_i\leqslant d\log\frac{\text{Tr}(M_n)}{d}\leqslant d\log(\lambda+Nc)$$

where $\sigma_i$ is the $i$-th eigenvalue of $M_n$. $\quad\square$

## F  EXPERIMENT DETAILS

### F.1  ONLINE SETTING

We list all the hyperparameter and network architecture we use for our experiments. For online MuJoCo and DM Control tasks, the hyperparameters can be found at Table 4. Therefore, we set bonus scaling term to 0 for MuJoCo tasks. However, this bonus is critical to the success of DM Control Suite (especially sparse reward environments). Note that we use exactly the same actor and critic network architecture for all the algorithms in the DM Control Suite experiment.

For evaluation in Mujoco, in each evaluation (every 5K steps) we test our algorithm for 10 episodes. We average the results over the last 4 evaluations and 4 random seeds. For Dreamer and Proto-RL, we change their network from CNN to 3-layer MLP and disable the image data augmentation part (since we test on the state space). We tried to tune some of their hyperparameter (e.g., exploration steps in Proto-RL) and report the best number across our runs. However, due to the short time, it is also possible that we didn't tune the hyperparameter enough.

Table 4: Hyperparameters used for SPEDER in all the environments in MuJoCo and DM Control Suite.

|  | Hyperparameter Value |
| --- | --- |
| C | 1.0 |
| regularization coef | 1.0 |
| Bonus Coefficient (MuJoCo) | 0.0 |
| Bonus Coefficient (DM Control) | 5.0 |
| Actor lr | 0.0003 |
| Model lr | 0.0003 |
| Actor Network Size (MuJoCo) | (256, 256) |
| Actor Network Size (DM Control) | (1024, 1024) |
| SVD Embedding Network Size (MuJoCo) | (1024, 1024, 1024) |
| SVD Embedding Network Size (DM Control) | (1024, 1024, 1024) |
| Critic Network Size (MuJoCo) | (1024, 1) |
| Critic Network Size (DM Control) | (1024, 1) |
| Discount | 0.99 |
| Target Update Tau | 0.005 |
| Model Update Tau | 0.005 |
| Batch Size | 256 |

### F.2  PERFORMANCE CURVES

We provide the performance curves for online DM Control Suite experiments in figure 1. As we can see in the figures, the proposed SPEDER converges faster and achieve the state-of-the-art performances in most of the environments, demonstrating the sample efficiency and the ability to balance of exploration vs. exploitation of SPEDER.

### F.3  TRANSITION ESTIMATION VIA SPECTRAL DECOMPOSITION

We show that the SPEDER objective can learn valid transitions of the environment. We use a empty-room maze environment, where the state is the position of the agent and the action is the velocity. The transition can be expressed as $s' = s + at + \epsilon$, where $t$ is a fixed time interval and $\epsilon \sim \mathcal{N}(0, I)$. We run SPEDER for $100K$ steps and the learned transition heatmap is visualized in Figure 2. The blue region is the heatmap estimation via spectral decomposition and S1 is the target position of the agent. The high density region is centered around the red dot (target

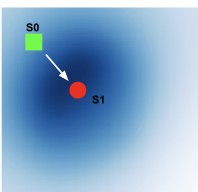

Figure 2: Estimated Transition via SPEDER.

position S1), which means the representation learned by our objective captures the environment transition. This shows the spectral decomposition can learn a good transition function.

### F.4  IMITATION LEARNING

For all methods, we use latent behavioral cloning as described in Section 3.2 to pre-train representations on a suboptimal dataset $\mathcal{D}^{\text{off}}$, then finetune on the expert dataset $\mathcal{D}^{\pi^*}$ for downstream imitation learning. We also compare with baseline behavioral cloning (BC) (Pomerleau, 1998), which directly learns a policy from the expert dataset (without latent representations) by maximizing the log-likelihood objective, $\mathbb{E}_{(s,a)\sim\text{Pr}(\mathcal{D}^{\pi^*})}\left[-\log\pi(a\mid s)\right]$.

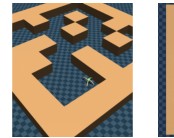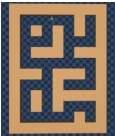

Figure 3: AntMaze navigation domains in mazes of medium (left) and large (right) sizes.

We report the average return on AntMaze tasks, and observe that SPEDER achieves comparable performance as other state-of-the-art representations on downstream imitation learning (Figure 4). We also observe that the normalized marginalization regularization equation 11 helps performance (Figure 5). We provide the performance curves for imitation learning in Figures 6 and 7.

For TRAIL, OPAL, SPiRL, SKiLD, and BC, we used the same hyperparameters as reported in Yang et al. (2021). For all methods, we pre-trained the representations for 200K steps using Adam optimizer with a learning rate of 3e-4 and batch size 256. For latent behavioral cloning, we train the latent policy $\pi_Z$ for 1M iterations using a learning rate of 1e-4 for BC, SPiRL, SKiLD, and OPAL, and 3e-5 for SPEDER and TRAIL (both EBM and Linear). We found that decaying the BC learning rate helped prevent overfitting for all methods. We evaluate the policy every 10K iterations by rolling out the policy in the environment for 10 episodes, and recording the average return. The representations $\phi$ and action decoder $\pi_\alpha$ were frozen during downstream behavioral cloning. All imitation learning results are reported over 4 seeds.

Both the action decoder $\pi_\alpha$ and the latent policy $\pi_Z$ are parameterized as a multivariate Gaussian distribution, with the mean and variance approximated using a two-layer MLP network with hidden layer size 256.

For SPEDER and TRAIL, $\phi$ and $\mu$ are parameterized as a 2-layer MLP with hidden layer size 256, and a Swish activation function (Ramachandran et al., 2017) at the end of each hidden layer. We ran a sweep of embedding dimensions $d \in \{64, 256\}$ and found that $d = 64$ worked best for TRAIL, and $d = 256$ worked best for SPEDER. For SPEDER, we ran a sweep of coefficients for each loss term in equation 10, and summarize the coefficients used in Table 5. For TRAIL Linear, we used a Fourier dimension of 8192, which has been provided more preference, while still performing worse.

For SPiRL, SKiLD and OPAL, we used an embedding dimension of 8, which was reported to work best (Yang et al., 2021). The trajectory encoder is parameterized as a bidirectional RNN, and the skill prior is parameterized as a Gaussian network following (Ajay et al., 2020). SPiRL and SKiLD are adapted for downstream behavioral cloning by minimizing the KL divergence between the latent policy and the skill prior.

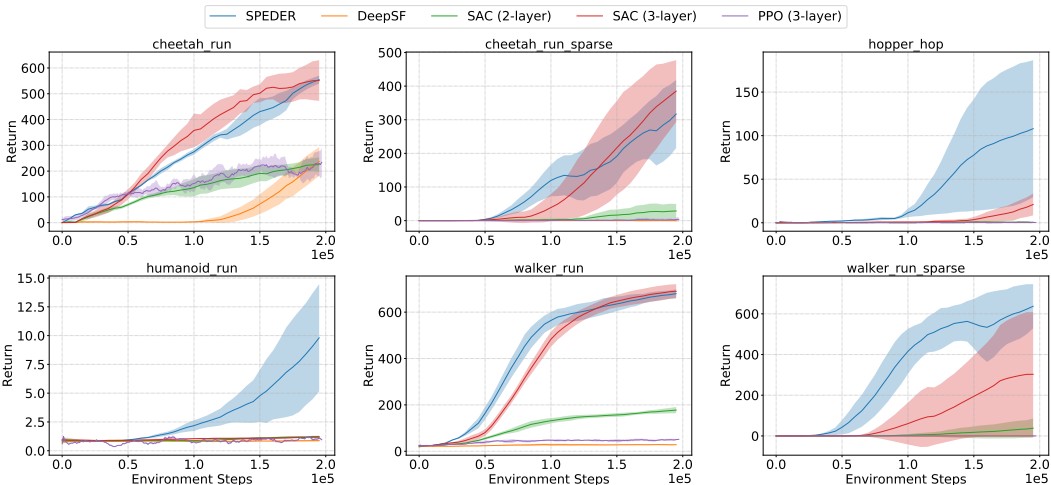

Figure 1: Performance Curves for online DM Control Suite.

Table 5: Loss coefficients used for SPEDER equation 10. We denote the loss coefficients as $a_1$ for the $\mathbb{E}_{p(s')}\left[\mu(s')^\top \mu(s')\right]/(2d)$ term; $a_2$ for the $\mathbb{E}_{(s,a)\sim\rho_0}\left[\phi(s,a)\phi(s,a)^\top\right] = I_d/d$ term; and $a_3$ for the additional normalization regularization term in equation 11.

| Domain | SPEDER w/o normalization | | SPEDER w/ normalization equation 11 | | |
|---|---|---|---|---|---|
| | $a_1$ | $a_2$ | $a_1$ | $a_2$ | $a_3$ |
| antmaze-large-diverse | 0.1 | 0.1 | 0.01 | 0.01 | 1. |
| antmaze-large-play | 0.01 | 0.01 | 1. | 0.01 | 1. |
| antmaze-medium-diverse | 1. | 0.01 | 0.1 | 1. | 0.1 |
| antmaze-medium-play | 1. | 0.01 | 1. | 0.1 | 1. |

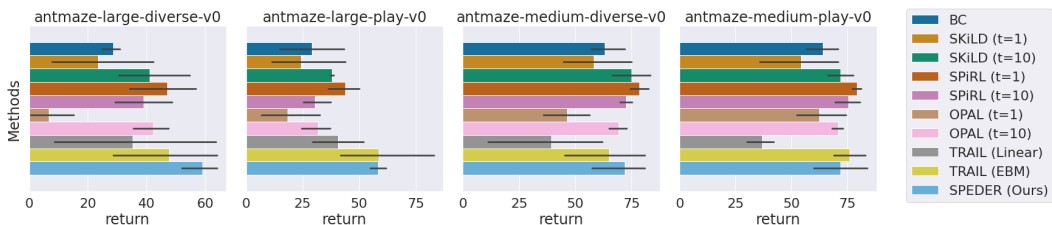

Figure 4: Average return on imitation learning tasks from D4RL AntMaze (Fu et al., 2020). BC corresponds to behavioral cloning on the expert dataset without latent representations. All other methods pre-train representations on a suboptimal dataset, and then finetune on an expert dataset.

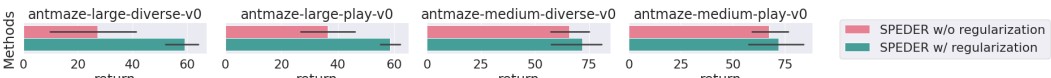

Figure 5: Ablation of SPEDER with vs. without normalized marginalization regularization equation 11.

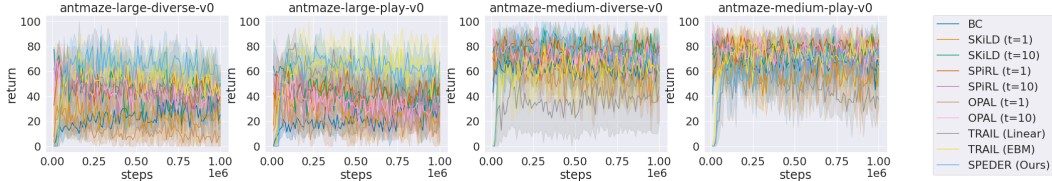

Figure 6: After pre-training, we train latent behavioral cloning on top of the learned representations for 1M iterations. BC refers to direct behavioral cloning on the expert dataset without latent representations. The corresponding barplot of the final performance is provided in Figure 4.

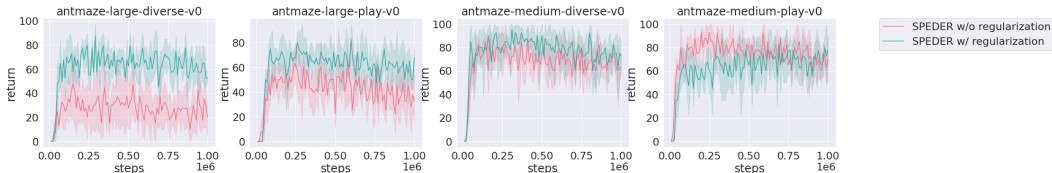

Figure 7: Performance curve of downstream behavioral cloning for SPEDER with vs. without normalized marginalization regularization equation 11. The corresponding barplot of the final performance is provided in Figure 5.

