# OpenReview forum: "Spectral Decomposition Representation for Reinforcement Learning"
_ICLR.cc/2023/Conference — ICLR 2023 poster_

### Official Review · Reviewer_PTm5 · 2022-10-24

**Confidence:** 2
**Correctness:** 4
**Technical Novelty And Significance:** 3
**Empirical Novelty And Significance:** 3
**Recommendation:** 8

**Clarity, Quality, Novelty And Reproducibility:**

The paper seems to contain interesting results.
The paper needs to be clarified more in order to improve the readability.
Overall, the paper is well written, and it clearly compares the proposed method with existing ones through numerical examples.

**Strength And Weaknesses:**

The paper seems to contain interesting results.
Some typos need to be corrected, for example, "sample" => "simple in Introduction.
The presentation can be modified in order for the reader who is not familier with representation learning.
For example, what is the difference between representatoin learning and RL with function approximation?

**Summary Of The Paper:**

The paper proposes a novel objective, Spectral Decomposition Representation (SPEDER), that factorizes the state-action transition kernel to obtain policy-independent spectral features. The authors show how to use the representations obtained with SPEDER to perform sample efficient online and offline RL, as well as imitation learning. Moreover, the paper provides a thorough theoretical analysis of SPEDER and empirical comparisons on multiple RL benchmarks, demonstrating the effectiveness of SPEDER.

**Summary Of The Review:**

The paper proposes a novel objective, Spectral Decomposition Representation (SPEDER), that factorizes the state-action transition kernel to obtain policy-independent spectral features.
Overall, the paper is well written, and it clearly compares the proposed method with existing ones through numerical examples.

---

> ### Author Response · Authors · 2022-11-15
> **Author Response to reviewer PTm5**
>
> We thank the reviewer PTm5 for the positive feedback and suggestions on the readability. We have revised our paper to improve the readability. We would like to provide the clarifications below:
>
> * ** What is the difference between representation learning and RL with function approximation**: Representation learning algorithms indeed use powerful function approximators to extract informative representations, so it is subject to the set of RL algorithms with function approximation. On the other hand, representation learning mainly focuses on constructing the appropriate function class from data based on some structural assumptions such as low-rank MDPs, with which we can obtain provably better sample complexity with practical empirical algorithms. This is in sharp contrast with the research on RL with function approximation, which generally relies on a given function space.

---

> > ### Comment · Reviewer_PTm5 · 2022-12-14
> > **Response to the authors's response**
> >
> > Thank you. The authors addressed my previous comments well. I would like to keep my previous score.

---

### Official Review · Reviewer_jSW9 · 2022-10-26

**Confidence:** 2
**Correctness:** 4
**Technical Novelty And Significance:** 4
**Empirical Novelty And Significance:** 3
**Recommendation:** 8

**Clarity, Quality, Novelty And Reproducibility:**

### Clarity
The paper is sufficiently clear given the density of the problems it tackles.

### Quality
The paper and research is of good quality. Some questions:

### Novelty
This is novel analysis, to the best of my knowledge.

### Reproducibility
Hyperparameters and experimental setup seem to have been provided. However, open sourced code would have been nice to see (I couldn't find any links in the paper).

**Strength And Weaknesses:**

### Strengths
- There is a good introduction to the spectral decomposition view.
- Good theoretical analyses.
- Fair literature review, with a description of why the present work is needed.
- Remark on model based variant of this work was interesting to see.
- Algorithmic steps are well clearly.


### Weaknesses
- The paper can be a little dense to read, but this could be because of the subject matter at hand.

**Summary Of The Paper:**

This paper provides a novel method of spectral representation learning that is compatible with stochastic gradient descent, allows for sample efficient online exploration, and

A key novelty is that inter-state dependence is avoided by extracting spectral features some a policy independent state transition operator.

The paper provides analyses generalization bounds and sample complexities of the method.


**Summary Of The Review:**

Based on my limited understanding of the related work in this line of thinking, I believe that this is good research. The background theory is well described, along with a good literature review. The algorithmic details of the method, as well as the theoretical analysis are presented well. I did not check the correctness of these analyses however.

---

> ### Author Response · Authors · 2022-11-15
> **Author Response to Reviewer jSW9**
>
> We thank the reviewer jSW9 for the overall positive feedback. We have revised our paper to improve the readability and we  will open-source our implementation after our manuscript gets accepted.

---

### Official Review · Reviewer_uUqZ · 2022-10-30

**Confidence:** 4
**Correctness:** 3
**Technical Novelty And Significance:** 2
**Empirical Novelty And Significance:** 2
**Recommendation:** 5

**Clarity, Quality, Novelty And Reproducibility:**

Clarity/Quality/Question:

1. Adding a pseudocode of the algorithm that was ran experimentally would improve clarity. At the least, the empirical version of Equation 10 that was tried should be included.

2. Adding more details for Equation 6 and Equation 7, and defining notations would also help clarity.

3. Please explain how adding Equation 11 ensures that $\hat{P}$ is a valid probability distribution?

4. How is planning performed? If $\hat{P}$ is not used for planning, as it may not be a valid distribution, then how is the replay dataset used to perform planning?

Novelty:
The paper is very close to prior work on low-rank MDPs such as RepUCB in how the data is collected and elliptic bonus-based planning is done. However, the proposed representation learning approach as given by Equation 10 is novel. Further, the paper includes experiments unlike prior work.


**Strength And Weaknesses:**

Strength:

1. Low-rank MDPs are widely studied and the present paper proposes a new approach that claims to be more computationally-efficient
2. Experiments and theoretical statements show promise of this approach

Weakness:

1. It is unclear if the proposed approach is more efficient than RepUCB. For example, RepUCB does not use partition function since the model class is assumed to be normalized. Is that not possible in practice? The objective proposed in the paper seems quite complex and hard to optimize. While experiments show that it can be optimized, certain gaps exists between theory and practice which makes it hard to understand to what extent, Algorithm 1 is tried empirically. E.g., Algorithm 1 says that policy is found by using model-based planning on $\hat{P}$, but the paragraph on the requirement on $\hat{P}$ suggests that planning is done somehow using the replay memory data. A pseudocode of the algorithm that was run experimentally would be helpful.

2. The paper does not present all details which makes it hard to verify certain claims. To begin with, the main contribution is the SVD approach described in Equations 6 and 7. However, the very first line uses a notation that is not defined. Specifically, what is meant by $\langle P(s' \mid s, a), \phi(s, a) \rangle_{\rho_0}$. Normally, one uses $\langle x, y \rangle_A$ to denote $x^\top A y$ which is an inner product if $A$ is a positive definite.  However, I am not sure how $\rho_0$ is represented as a $S \times A$ matrix. Is it diagonal with $\rho(s, a)$ as diagonal entries? Similarly, the explanation of Equation 7 can be further expanded.

3. Lastly, even though the paper aims to improve upon algorithms like RepUCB, FLAMBE, and MOFFLE; not even one of these algorithms is experimentally tried. If these algorithms are impossible to try experimentally, then the paper should discuss why this is the case. Otherwise, including their performance will help understand the effectiveness of SPEDER.

**Summary Of The Paper:**

This paper studies representation learning in low-rank MDPs. In low-rank MDPs, the transition model $T(s' \mid s, a)$ can be expressed as a low-rank decomposition $T(s' \mid s, a) = \mu(s')^\top \phi(s, a)$ and similarly the reward model $R(s, a) = \theta^\top \phi(s, a)$. The proposed approach SPEDER takes an SVD approach to learn the representation $\phi$ as opposed to performing maximum-likelihood estimation used in prior work. The paper claims that this leads to a simpler optimization. Once $\phi$ is learned, the approach closely mimics existing work in using elliptic bonus, and planning a policy using the learned model with bonus.

**Summary Of The Review:**

I am currently leaning towards a tentative score of weak reject due to a lack of clarity about the algorithm that was empirically run, and questions regarding both empirical and conceptual differences between RepUCB and SPEDER. Including empirical performance of RepUCB, or adding clarity on why Equation 10 is easier to optimize than the MLE objective in RepUCB, would most influence the score.

---

> ### Author Response · Authors · 2022-11-15
> **Author Response to Reviewer uUqZ (2/2)**
>
> We make the following revisions based on the suggestions:
>
> * Implementation Details: We provide additional implementation details in Appendix B.
> * Revision of Section 3: We have updated the manuscript to eliminate the potential ambiguities in Section 3 as suggested by reviewer uUqZ.
>
> We hope the response and the revised manuscript can address your concern. Please feel free to reach out to us if there are any remaining questions.

---

> ### Author Response · Authors · 2022-11-15
> **Author Response to Reviewer uUqZ (1/2)**
>
> We thank the reviewer uUqZ for the detailed comments and suggestions to improve the paper. We would like to address the concern below.
>
> * **Is RepUCB possible in practice?**: When the state space is combinatorial or infinite, which is common in practical control applications, we cannot efficiently compute the exact partition function, and we are not aware of any parameterization method to make the parameterized conditional density self-normalized. In this case, to perform conditional MLE, we need to estimate the partition function to certain accuracy, which can be extremely difficult. It is also unclear what will happen if we replace the exact partition function with the corresponding estimate. Hence, we believe RepUCB is not easy to be implemented in most real world applications and we need to consider alternative methods for representation learning, as we have discussed in Sec 3.
>
> * **Comparison of our objective and conditional MLE**: Motivated by the aforementioned difficulties in the implementation of conditional MLE,  we propose a novel objective that can be estimated with standard stochastic approximation. Ignoring the orthogonal constraints for now, for the objective in Equation 9, all of the terms can be estimated through standard Monte-Carlo estimation, which can be more favorable for *practical use cases*. We can further add the orthogonality constraint, to obtain more informative representations.
> As we mentioned in the manuscript, Equation 10 is equivalent to Equation 7, and Equation 7 can be handled via the penalty method as in [1]. Specifically, we can introduce a regularization term on the correlation between different coordinates of $\phi$ and use any gradient-based optimization algorithm to maximize the objective function.
>
> * **How to perform planning**: We perform planning with the soft actor-critic (SAC) algorithm, where the critic is parameterized with the learned representation, and optimized with the data in the replay buffer using the Fitted Q-iteration (FQI) algorithm. We re-emphasize this in the latest manuscript. We would like to further remark that the requirement that $\hat{P}$ is a probability is mainly used for theoretical analysis. In practice, this method can already work well, even if we don't add the additional regularization in Equation 11. The understanding of such phenomenon is subject to further study, and is beyond the scope of the current manuscript.  We added the implementation details in Appendix B.
>
> * **The notation of $\langle P(s^\prime|s, a), \phi(s, a)\rangle_{\rho_0}$**: We are sorry for the confusion here. Here we want to use this term to represent $\mathbb{E}_{(s, a) \sim \rho_0} \langle P(s^\prime|s, a), \phi(s, a)\rangle := \int P(s^\prime|s, a)\phi(s, a) d\rho_0$ where the inner product is in the $L_2(\mu)$ space where $\mu$ is a proper measure (Lebesgue measure for continuous case and counting measure for discrete case) on the state-action space. To further make the notation consistent with the context, we have revised Section 3 in the latest manuscript.
>
> * **The Effect of Equation 11**: As our target is $P(s^\prime|s, a) = \phi(s, a)^\top p(s^\prime)\mu(s^\prime)$, we can guarantee $P(s^\prime|s, a) \geq 0$ by forcing each coordinate of $\phi(s, a)$ and $\mu(s^\prime)$ to be larger than or equal to $0$. This can be achieved by our parameterization, for example, applying a ReLU or sigmoid activation function at the end of the forward pass of the neural network. Furthermore, we want $\int_{\mathcal{S}}  \phi(s, a)^\top p(s^\prime)\mu(s^\prime) d s^\prime = 1$. As the minimum of $(\log x)^2$ is achieved when $x = 1$, adding such regularization can help enforce that $\hat{P}$ is a probability. This method has also been used in e.g. [2]. We have added a sentence to clarify this in the latest manuscript.
>
> * **Empirical Comparison with RepUCB, FLAMBE and MOFFLE**: The aforementioned algorithms are mainly designed for theoretical analysis and can be extremely difficult for practical implementation. As far as we know, there is no practical implementation for any of these algorithms and no empirical study has been conducted, which motivates our proposed method.
> As we mentioned, RepUCB assumes an MLE oracle for the computation, but to the best of our knowledge, there are no efficient algorithms to conduct MLE for conditional density estimation under such parameterization when the event space is continuous. FLAMBE also requires an MLE oracle, and it further uses a policy-cover based method for exploration, that requires keeping **large amounts of policies** in memory, which can be memory-inefficient. MOFFLE instead requires solving a *min-max-min* problem for the representation learning, which can also be hard since the optimization landscape can be non-convex and non-concave.

---

### Decision · Program_Chairs · 2023-01-20

**Decision:**

Accept: poster

**Justification For Why Not Higher Score:**

See the concern (B) above.

**Justification For Why Not Lower Score:**

See the reason (A) above.

**Metareview: Summary, Strengths And Weaknesses:**

The main contribution of this work lies in proposing the Spectral Decomposition Representation that can extract a state-action abstraction from the dynamics to obtain policy-independent spectral features. This work provides rigorous theoretical analyses and sufficiently extensive empirical evaluation.

After reviewing the authors' rebuttal, the reviewers agree that (A) the contribution of this work is novel and interesting.

The reviewers have also raised (B) the concern of readability/clarity of presentation, especially from Reviewer uUqZ. The authors have since revised their manuscript to address the reviewers' concern adequately.

**Note From Pc:**

if the above contains the word "oral" or "spotlight" please see: "oral" presentation means -> notable-top-5% and "spotlight" means -> notable-top-25%. As stated in our emails, we are disassociating presentation type from AC recommendations